# THINK-AT-HARD: SELECTIVE LATENT ITERATIONS TO IMPROVE REASONING LANGUAGE MODELS

## ABSTRACT

Improving reasoning capabilities of Large Language Models (LLMs), especially under parameter constraints, is crucial for real-world applications. Prior work proposes recurrent transformers, which allocate a fixed number of extra iterations per token to improve generation quality. After the first, standard forward pass, instead of verbalization, last-layer hidden states are fed back as inputs for additional iterations to refine token predictions. Yet we identify a *latent overthinking* phenomenon: easy token predictions that are already correct after the first pass are sometimes revised into errors in additional iterations. To address this, we propose Think-at-Hard (TaH), a dynamic latent thinking method that iterates deeper only at hard tokens. It employs a lightweight neural decider to trigger latent iterations, only at tokens that are likely incorrect after the standard forward pass. During latent iterations, Low-Rank Adaptation (LoRA) modules shift the LLM's objective from general next-token prediction to focused hard-token refinement. We further introduce a duo-causal attention mechanism that extends attention from token sequence dimension to an additional iteration depth dimension. This enables cross-iteration information flow while maintaining full sequential parallelism. Experiments show that TaH boosts LLM reasoning performance across five challenging benchmarks while maintaining the same parameter count. Compared with baselines that iterate twice for all output tokens, TaH delivers 8.1-11.3% accuracy gains while exempting 94% of tokens from the second iteration. Against strong single-iteration Qwen3 models finetuned with the same data, it also delivers 4.0-5.0% accuracy gains. When allowing <3% additional parameters from LoRA and iteration decider, the gains increase to 8.5-12.6% and 5.3-5.4%, respectively.

## 1 INTRODUCTION

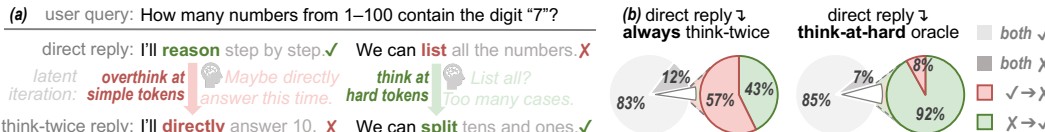

Figure 1: Selective iteration can mitigate latent overthinking. (a) Toy example. Uniform latent iteration (always think-twice) can fix wrong predictions, but may also overthink and corrupt correct ones. (b) Next-token prediction accuracy of finetuned Qwen3-1.7B variants. Always think-twice causes more errors than corrections over direct reply. In contrast, the think-at-hard oracle, which iterates only when the first-pass prediction is wrong, achieves substantial improvements with minimal harm. While this oracle signal is unavailable in practice, it highlights the potential of selective iteration.

Recent advances in Large Language Model (LLM) reasoning have enabled broad applications across diverse domains (Jaech et al., 2024; Guo et al., 2025; Yang et al., 2025). With tens to hundreds of billions of parameters, LLMs can generate complex Chain-of-Thought (CoT) to solve challenging tasks. At the same time, smaller language models have also drawn increasing attention. With only a few billion parameters, they offer compelling alternatives: lower costs, faster inference, and suitability for edge computing (Abdin et al., 2024; Team et al., 2025; Wang et al., 2025a).

At this crossroad, enhancing reasoning capabilities under parameter constraints becomes a central challenge. A common approach is to distill smaller models to mimic LLM CoT trajectories using next-token prediction supervision. However, not all tokens are equally predictable: certain tokens encode critical logic or reasoning directions that are fundamentally harder to predict (Lin et al., 2024; Fu et al., 2025a; Wang et al., 2025b). With limited computation per output token, small models quickly hit a performance ceiling and mispredict some of these tokens. Once critical errors occur, the reasoning trajectory can irrecoverably diverge and produce drastically different outcomes.

Prior work proposes recurrent transformers to address this parameter–performance paradox (Hutchins et al., 2022; Saunshi et al., 2025; Zeng et al., 2025). Instead of verbalizing the next token immediately after one forward pass, these models typically feed the last-layer hidden states back into the LLM for additional passes in the latent space. Each pass refines the hidden representation without producing tokens. After a fixed number of iteration depths, the final hidden states are passed to the language modeling head to generate the next token. By uniformly allocating extra iterations per token, these models increase inference depth without enlarging parameter count, potentially benefiting hard reasoning tokens.

However, we identify a *latent overthinking* problem in fixed-depth recurrent transformers, where excessive iterations revise correct answers into wrong ones. As shown in Figure 1, finetuning Qwen3-1.7B-Base to always perform two iterations per token yields even more errors than the single-iteration baseline on the Open-R1 dataset (Hugging Face, 2025). This occurs because most tokens are already predicted correctly in the first iteration, such as coherence or suffix tokens. Similar to overthinking in explicit CoT reasoning (Wu et al., 2025), latent overthinking on these easy tokens degrades performance despite extra computation. While the opposite *latent underthinking* exists for tokens that need more iterations to correctly predict, such cases are rarer. We define tokens that cannot be accurately predicted in a single forward pass as *hard* tokens, and ask our central question:

*Can LLMs selectively dedicate latent iterations only to hard tokens?*

If achieved, different iterations could specialize in distinct prediction focuses for more effective latent reasoning. Oracle experiments validate this approach: as shown in Table 4, a think-at-hard oracle improves MATH accuracy by 25-28%.

Achieving dynamic latent iteration presents three main challenges. First, the model architecture should enable cross-depth attention, allowing each iteration to access full context. This is crucial because when early tokens skip deeper iterations, later tokens must still access their representations from shallower depths. Meanwhile, this cross-depth flow cannot compromise the sequence-level parallelism essential for efficient training and prefilling. Second, the model must adapt to changing objectives and distributions across iterations, while maximizing parameter reuse. Third, training must remain stable despite tight coupling dependencies: the iteration policy depends on prediction quality at each depth, while that quality depends on which tokens the policy sends to each depth.

To address these challenges, we propose TaH, a dynamic latent thinking method that selectively applies deeper iterations only to hard tokens. As shown in Figure 2, TaH employs a neural decider to determine whether to continue iterating or verbalize each token. We design a duo-causal attention mechanism to enable cross-depth attention and full sequence parallelism. To specialize deeper iterations for hard-token refinement and preserve strong first-pass predictions, we apply LoRA adapters solely at iterations $d > 1$. TaH is stably trained by aligning both LLM backbone and iteration decider with a static oracle iteration policy. We summarize our contributions as follows.

- **Selective Latent Iteration**. We identify the latent overthinking phenomenon, revealing how false corrections harm easy tokens at redundant iterations. This insight guides our new paradigm where latent iteration depth adapts to token difficulty.

- **Specialized Model Architecture**. We develop a model architecture that natively supports selective iteration depths. The dedicated duo-causal attention mechanism, LoRA adapters, and iteration decider enable efficient cross-depth information flow, objective transitions, and dynamic depth selection.

- **Stable Training**. We introduce a stable training scheme that uses a static oracle policy to decouple model adaptation and policy learning. It overcomes the circular dependency between iteration decisions and prediction quality.

Experiments show that TaH consistently improves reasoning performance. Finetuned from Qwen3-0.6B-Base and 1.7B-Base with aligned parameter count, TaH achieves an average accuracy gain of 4.0-5.0% over standard single-iteration variants across five reasoning benchmarks, while applying deeper thinking to only 6% of tokens. With less than 3% additional parameters, these gains further increase to 5.3-5.4%. Compared with AlwaysThink which applies two iterations to all tokens, the gains are 8.1-11.3% and 8.5-12.6%, validating TaH's high effectiveness.

## 2  RELATED WORK

Unlike standard LLMs that verbalize at every autoregressive step, latent thinking shifts part of generation away from explicit natural-language CoT in order to improve reasoning (Li et al., 2025).

**Signal-guided Control**. These methods keep reasoning in token space but steers computation by inserting control tokens. Early work shows that simple filler tokens (e.g., dots) can mimic some benefits of CoT (Pfau et al., 2024). Building on this, later work expands the LLM vocabulary with `[PAUSE]` tokens and learns where to insert them for extra compute before predicting the next token (Goyal et al., 2024; Kim et al., 2025). They are lightweight and easily integrable, but constrained to the discrete-token interventions with limited latent controls.

**Latent Optimization**. These methods perform autoregressive reasoning directly in internal representations, emitting little or no intermediate text. They distill and compress CoT into latent continuous embeddings through various strategies. Coconut and CCoT progressively replace text with latent thinking under final response supervision (Hao et al., 2024; Cheng & Van Durme, 2024); Token assorted and HCoT compress CoT spans to embeddings with hidden-state alignment (Su et al., 2025; Liu et al., 2024). SoftThink directly applies logit-weighted embeddings for latent iterations (Zhang et al., 2025b). While offering efficiency gains and flexible control over hidden trajectories, these methods sacrifice reasoning interpretability, with training-based ones further requiring heavy mitigation from strong verbal LLMs.

**Recurrent Transformers**. These methods interleave latent and verbal reasoning, introducing latent iterations before each token verbalization. After a standard forward pass, these methods feed latent states back as next-iteration inputs for a fixed number of iterations, then verbalize the output token. Existing approaches differ in the formation of next-iteration input. For example, Looped Transformer reuses last-layer hidden states directly (Saunshi et al., 2025; Geiping et al., 2025), whereas Ponder uses logit-weighted embeddings (Equation 4) (Zeng et al., 2025). Recurrent transformers combine advantages of visible reasoning trajectories with latent exploration. By reusing the parameters across iterations, it achieves deeper computation per token without parameter increases. However, the fixed depth burdens each iteration with both easy and hard tokens, potentially causing false corrections for already-correct predictions.

**Positioning**. TaH belongs to the recurrent transformer family but extends this paradigm significantly. It *selectively* allocates latent iterations to *refine hard tokens*, improving reasoning quality with specialized objectives across iterations. While concurrent works (Bae et al., 2025; Zhu et al., 2025) also enable selective recursion, they require complete model retraining. TaH instead leverages existing pre-trained models, adding depth-aware LoRA and duo-causal attention to improve reasoning with minimal finetuning overhead.

## 3  PRELIMINARY

**Autoregressive LLMs**. Modern LLMs generate text through an autoregressive next-token prediction process. It includes a *prefill* stage and a *decode* stage (Radford et al., 2018; 2019; Kwon et al., 2023). In the prefill stage, the model processes the entire input sequence in parallel; in the decode stage, it consumes one new token at a time along with cached history to predict the next token.

Formally, let $t_i$ denote the token at position $i$ and $x_i \in \mathbb{R}^h$ its embedding. Let $E \in \mathbb{R}^{v \times h}$ be the embedding matrix, so $x_i = E[t_i]$ when $t_i$ is treated as an index. Here, $v$ and $h$ are the vocabulary size and hidden dimension. The output projection matrix is $W_{\text{out}} \in \mathbb{R}^{h \times v}$ (equal to $E^\top$ if tied). Given the context $T_{\leq i} = [t_0, \ldots, t_i]$ with embeddings $X_{\leq i} = [x_0, \ldots, x_i]$, the model $\theta$ produces a

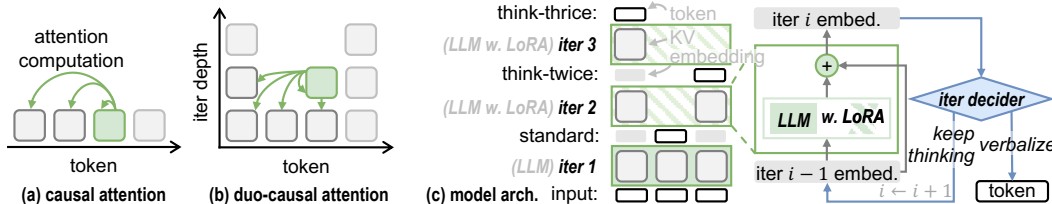

Figure 2: TaH Overview. (a) Regular causal attention: tokens attend only to previous positions. (b) Our duo-causal attention: tokens attend to both previous positions and shallower iteration depths, maintaining 2D causality. (c) Model architecture: TaH selectively iterates or verbalizes tokens. It uses LoRA at deeper iterations to shift from next-token prediction to hard-token refinement. A neural decider determines whether to continue iterating or output the token.

*last-layer hidden state* $y_i$ for token $t_i$:

$$y_i \;=\; \mathcal{P}_\theta\big(x_i \mid X_{\leq i}\big) \;\in\; \mathbb{R}^h. \tag{1}$$

The next-token distribution $p_i$ and sample are:

$$p_i \;=\; \mathrm{softmax}\big(W_{\mathrm{out}}^\top y_i\big) \in \mathbb{R}^v, \qquad t_{i+1} \;=\; \mathcal{S}(p_i), \tag{2}$$

where $\mathcal{S}$ is a sampling rule such as greedy or nucleus sampling. Decoding repeats until an end-of-sequence token is generated.

**Causal Attention**. To respect autoregression, modern LLMs apply *causal* attention. As shown in Figure 2(a), each position attends only to itself and earlier positions, consistent with Equation 1. This design brings two key benefits: (1) it enables parallel training with next-token prediction and shifted logits, avoiding the need for token-by-token generation; and (2) it allows efficient inference by caching Key/Value states of past tokens instead of recomputing them.

**Recurrent Transformers**. Recurrent transformers introduce an inner loop that iterates in latent space before verbalizing each output token. Let $d \in \{1, 2, \dots\}$ denote the iteration depth (written as a superscript), and set $x_i^{(0)} = E[t_i]$. At each iteration, recurrent transformers update $y_i$ with causal attention on the hidden states of *the current iteration*:

$$y_i^{(d)} \;=\; \mathcal{P}_\theta\big(x_i^{(d)} \mid X_{\leq i}^{(d)}\big), \qquad X_{\leq i}^{(d)} = [x_0^{(d)}, \dots, x_i^{(d)}]. \tag{3}$$

An inner transition then produces the next-depth embedding. For example, Loop (Saunshi et al., 2025) simply sets $x_i^{(d+1)} = y_i^{(d)}$, while Ponder (Zeng et al., 2025) uses a logit-weighted embeddings:

$$x_i^{(d+1)} = \mathrm{softmax}\big(W_{\mathrm{out}}^\top y_i^{(d)}\big)\, E = p_i^{(d)} E. \tag{4}$$

In practice, it uses the top-100 logits instead of full logits for efficiency.

Verbalization occurs at a fixed *maximum depth* $d_{\max}$ shared by all tokens, where $y_i^{(d_{\max})}$ is transformed into the next token $t_{i+1}$, resembling Equation 2.

## 4 TaH Design

We expand the motivations and key designs of TaH in this section, including the duo-causal attention mechanism (Section 4.1), model architecture (Section 4.2), and training scheme (Section 4.3).

### 4.1 Duo-Causal Attention

**Motivation**. In recurrent transformers, attention typically operates within each iteration. For fixed-depth methods, standard causal attention on the current iteration's Key and Value states already incorporates all context (Equation 3). However, dynamic iteration depths pose a challenge: tokens iterating at a deeper level cannot access the hidden states of previous tokens that verbalized at shallower depths. This creates a dilemma. On one hand, tokens require up-to-date states of all

previous tokens for rich semantic context. On the other hand, efficient training requires all tokens at depth $d$ be computable in parallel, without depending on previous tokens' deeper states ($d' > d$) that have not yet been computed. Existing approaches compromise on one of these aspects. Some sacrifice parallelism by allowing attention to deeper iterations, forcing sequential generation during training (Hao et al., 2024); others preserve parallelism by restricting attention to only the initial iteration's KVs (Bae et al., 2025). To resolve this dilemma, we introduce a simple yet effective mechanism to maximize cross-depth information flow while maintaining high parallelism.

**Duo-causal Attention Mechanism**. As shown in Figure 2(b), duo-causal attention extends *causality* to two dimensions, letting tokens attend across both previous positions and shallower iteration depths. Formally, we extend the accessible set from Equation 3 to

$$X_{\leq i}^{(\leq d)} \;=\; \{\, x_j^{(k)} \mid j \leq i, \; k \leq d \,\}. \tag{5}$$

When all tokens iterate only once (as in standard transformers), this naturally reduces to regular causal attention. The duo-causal design achieves both full parallel training and cross-depth information flow. At depth $d$, all tokens compute their depth-$d$ representations simultaneously using *only and all* information from depths 1 through $d$.

Implementation-wise, duo-causal attention is fully compatible with attention kernels like FlashAttention (Dao et al., 2022; Dao, 2024; Shah et al., 2024), or other sparse implementations (Fu et al., 2025b; Zhang et al., 2025a). As detailed in Appendix A.4.1, we simply maintain separate KV caches per iteration depth and flatten the 2D (token, depth) grid into a 1D sequence by concatenating deeper KV caches after shallower ones (Figure 14). Positional encodings are applied based solely on the original token index, invariant to iteration depth. The duo-causal constraint is then enforced via a modified additive attention mask, requiring no custom CUDA kernels.

## 4.2 MODEL ARCHITECTURE

**Motivation**. Previous fixed-depth recurrent transformers use identical weights across all iterations. However, we find that over 85% of next-tokens are correctly predicted at the first iteration (Figure 1(b)) This suggests deeper iterations serve a different objective: they refine the first iteration's prediction rather than predicting further ahead to the next-next token. This mirrors deep LLMs, where shallow layers predict next tokens for deeper layers to refine (Belrose et al., 2023; Schuster et al., 2022; Bae et al., 2023). While deep LLMs naturally handle this shift through distinct parameters per depth, recurrent transformers must accommodate both objectives with shared weights, potentially limiting performance. Moreover, fixed iteration depths can cause *latent overthinking*, motivating our dynamic approach.

**Backbone Model**. To address the objective shift, we apply a LoRA adapter (Hu et al., 2022) to the shared LLM backbone only for iterations $d > 1$. As shown in Figure 2(c), this allows the base LLM to focus on latent embeddings, while the adapter handles the objective shift. It preserves strong next-token prediction at $d = 1$, alleviating interference from deeper iterations. We also add residual connections across iterations to simplify the refinement and improve gradient flow. Formally, at depth $d$, we compute

$$y_i^{(d)} \;=\; \mathcal{P}_{\theta_d}\!\Big(x_i^{(d)} \,\Big|\, X_{\leq i}^{(\leq d)}\Big), \tag{6}$$

with depth-specific parameters

$$\theta_d = \theta \text{ for } d = 1, \qquad \theta_d = \theta + \Delta \text{ for } d > 1,$$

where $\theta$ and $\Delta$ denote the LLM and LoRA weights, respectively. The next-iteration inputs use logit-weighted embeddings (Equation 4); verbalization follows standard sampling (Equation 2). Each $y_i^{(d)}$ either continues iterating or verbalizes according to the decider $\mathcal{I}_\phi$.

**Iteration Decider**. We use a lightweight MLP as the iteration decider $\mathcal{I}_\phi$ to determine whether each token should continue iterating or verbalize. After each iteration, it processes concatenated hidden states from shallow, middle, and final layers of the backbone LLM to predict a continuation probability:

$$\hat{c}_i^{(d)} \;=\; \mathcal{I}_\phi\big(h_i^{(d)}\big) \in [0, 1]. \tag{}$$

During inference, token $i$ verbalizes when $c_i^{(d)}$ falls below threshold $c_{\text{threshold}}$ or reaches maximum depth $d_{\max}$.

## 4.3 TRAINING SCHEME

We employ a two-stage training scheme: first finetune the backbone model for dynamic iteration, then the iteration decider, all using an oracle policy.

**Motivation**. As shown in Figure 2(c), the backbone network $\theta_d$ and the neural iteration decider $\mathcal{I}_\phi$ are tightly coupled: the backbone generates hidden states as inputs for the decider, while the decider controls the backbone's KV cache and iterations. Training both simultaneously causes instability due to mutual distribution shifts. Therefore, we adopt a stable two-stage approach where both components are sequentially trained to align an oracle iteration policy $\pi$.

**Oracle Iteration Policy** $\pi$. To guide training, we define an oracle policy $\pi$ that determines token difficulty using a frozen reference LLM, following Fu et al. (2025a). A token is classified as *easy* if the reference model correctly predicts it with a single forward pass, and *hard* otherwise. Throughout the paper, we use the supervised fine-tuned (SFT) variant of the base model as the reference model.

Formally, let $\hat{t}_{i+1}$ denote the reference model's top-1 prediction and $t_{i+1}$ the ground-truth token. For explanation simplicity, we assume maximum iteration depth $d_{\max} = 2$ in Equation 7; the general case is detailed in Appendix A.2.4. The oracle iteration depth $d^\pi$ is:

$$d_i^\pi \ = \ 1 + \mathbf{1}\big[\hat{t}_{i+1} \neq t_{i+1}\big], \tag{7}$$

where $\mathbf{1}[\cdot]$ is the indicator function. The per-depth continuation label becomes:

$$c_i^{(d)} \ = \ \mathbf{1}[d \leq d_i^\pi], \tag{8}$$

indicating whether iteration should continue at depth $d$. Table 4 and Figure 1 verify the effectiveness of the oracle policy.

**Stage 1: Backbone supervision under $\pi$.** We optimize the backbone LLM ($\theta$ and LoRA adapter $\Delta$) with $\pi$-guided iteration execution. The loss is standard next-token prediction at the oracle-determined depth:

$$\mathcal{L}_{\text{SFT}}(\theta, \Delta) \ = \ \sum_i -\log p_i^{(d_i^\pi)}(t_{i+1}),$$

where $p_i^{(d_i^\pi)}$ is the next-token distribution at position $i$, depth $d_i^\pi$. This preserves first-iteration accuracy for easy tokens while training deeper iterations to refine hard tokens.

**Stage 2: Decider imitation under frozen backbone.** We freeze the backbone model ($\theta, \Delta$) and train the iteration decider $\phi$ to imitate the oracle policy's continuation decisions. We minimize binary cross-entropy with class reweighting for label imbalance:

$$\mathcal{L}_{\text{dec}}(\phi) \ = \ -\sum_i \sum_{d=1}^{\min\{d_{\max}-1, d_i^\pi\}} \Big[ w_{\text{stop/cont.}}^{(d)} c_i^{(d)} \log \hat{c}_i^{(d)} + \big(1 - c_i^{(d)}\big) \log \big(1 - \hat{c}_i^{(d)}\big) \Big],$$

where $c_i^{(d)}$ is the ground-truth continuation label, $\hat{c}_i^{(d)}$ is the predicted probability, and $w_{\text{stop/cont.}}^{(d)}$ is the occurrence ratio of stop label divided by continue label, respectively.

Our two-stage scheme stabilizes training by decoupling backbone learning (conditioned on a fixed $\pi$) from policy learning (imitation of $\pi$).

## 5 EXPERIMENT

### 5.1 SETUP

We present key configurations here, with more detailed setups in the Appendix.

**Baselines**. We compare diverse methods under equal parameter budgets, using Qwen3-0.6B-Base and Qwen3-1.7B-Base (Yang et al., 2025) as backbones. We compare TaH over several fixed-depth strategies: (1) *Standard*, which always verbalizes directly and reduces to the original Qwen model; (2) *AlwaysThink*, which applies the maximum number of latent iterations to all tokens; (3) *SoftThink*,

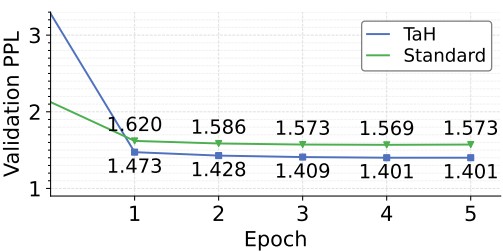 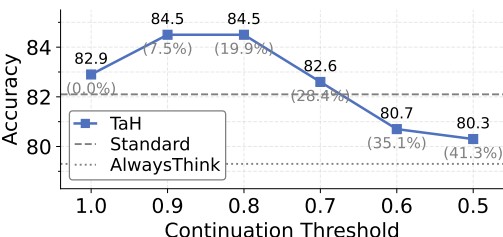

Figure 3: Training dynamics of the LLM backbone on Qwen3-0.6B-Base. TaH converges rapidly and achieves lower perplexity.

Figure 4: GSM8K accuracy with respect to continuation threshold. Numbers in brackets indicate the percentage of tokens that iterate twice.

following official baseline implementation (Zhang et al., 2025b) on top of the Standard model. Unless otherwise specified, both TaH and AlwaysThink use a maximum of two iterations. We also compare with dynamic query routing via matrix factorization (Ong et al., 2024), routing between MobileLLM-R1-360M (Zhao et al., 2025) and Qwen3-1.7B, as well as between Qwen3-0.6B and Qwen3-4B, to match average parameter sizes of 0.6B and 1.7B.

**TaH Setup**. Before training, we prune one layer from the base model so that TaH matches the parameter count of baselines. The layer is chosen to minimize the increase in validation loss. We also report results for an unpruned variant, TaH+, which adds less than 3% extra parameters from LoRA and iteration decider. The detailed parameter composition is shown in Table 6. Following (Fu et al., 2025a), we set the continuation threshold $c_{\text{threshold}} = 0.9$ with about 6% of tokens being iterated twice. The oracle policy $\pi$ uses Qwen3-0.6B, 1.7B and 4B as reference models to determine token difficulty during training.

**Training Scheme**. All models are trained on the math subset of Open-R1 (Hugging Face, 2025) using supervised finetuning. To fit memory and compute limits, we exclude responses longer than 8,192 tokens; 4B models additionally truncate at 4,096 tokens; all other training settings follow the official Open-R1 script. The filtered dataset contains 300M tokens, with 1% reserved for validation. Each method is sufficiently trained for 5 epochs, and we select the checkpoint with the lowest validation loss as the final model. All backbones are initialized from the corresponding Qwen3-Base.

**Evaluation Setup**. We evaluate across challenging reasoning benchmarks, including GSM8K (Cobbe et al., 2021), MATH500 (Hendrycks et al., 2021), AMC23 (American Mathematics Competitions), AIME25 (American Invitational Mathematics Examination), and Olympiad-Bench (He et al., 2024). The maximum generation length is set to 8,192 tokens for all benchmarks, except GSM8K which uses 4,096 due to its simpler problems and larger size. Performance is reported as pass@1 under a zero-shot chain-of-thought setting, using sampling temperature 0.6. For large datasets (MATH500, OlympiadBench, GSM8K), we generate one sample per problem; for small datasets (AMC23, AIME25), we generate eight samples per problem.

## 5.2 PERFORMANCE

**Benchmark Evaluation**. We validate TaH's reasoning ability through extensive tests across five challenging math benchmarks. Table 1 presents performance results for models at 0.6B and 1.7B parameter sizes. Starting from strong Qwen3-Base models, we observe that existing approaches show limited effectiveness: fixed-depth recurrent transformers (AlwaysThink) and query routing fail to consistently outperform the standard direct-answer baseline. SoftThink provides improvements on some cases, yet remain marginal overall. In contrast, TaH achieves consistent gains, delivering average improvements of 4.0% and 5.0% for the 0.6B and 1.7B models, respectively. Our enhanced variant (TaH+), which only adds less than 3% additional parameters, pushes these gains to 5.3% and 5.4%. Relative to AlwaysThink, the gains are 8.1-11.3% for TaH, and 8.5-12.6% for TaH+.

**Training Dynamics**. During stage 1 (LLM backbone training), guided by the oracle policy that only triggers a second iteration on hard tokens, TaH converges notably faster than the Standard baseline. It also achieves much lower perplexity on the validation dataset as shown in Figure 3. During stage 2

Table 1: Accuracy comparison of different baselines across five benchmarks and two model sizes. Subscripts indicate improvement over Standard. The top two scores for each task and model size are highlighted in bold.

| Param. | Benchmark | Method | | | | | |
|---|---|---|---|---|---|---|---|
| | | Standard | Routing | SoftThink | AlwaysThink | TaH | TaH+ |
| 0.6B | AIME25 | **4.2** | 1.0 | 2.5 | 1.5 | **4.2** | **5.0** |
| | OlympiadBench | 18.8 | 7.4 | 19.4 | 10.2 | **23.9** | **24.0** |
| | AMC23 | 23.4 | 10.9 | 24.1 | 15.6 | **32.5** | **30.6** |
| | MATH500 | 47.2 | 27.3 | 48.8 | 32.8 | **51.2** | **54.2** |
| | GSM8K | 62.5 | 45.6 | 61.3 | 54.6 | **64.4** | **68.8** |
| | Average | 31.2 | 18.5 | 31.2 | 22.9 | **35.2**$_{/+4.0}$ | **36.5**$_{/+5.3}$ |
| 1.7B | AIME25 | 13.3 | 10.2 | 12.9 | 10.0 | **17.9** | **14.6** |
| | OlympiadBench | 33.0 | 30.6 | 33.4 | 30.0 | **38.8** | **41.2** |
| | AMC23 | 42.2 | 42.2 | 43.1 | 42.5 | **48.4** | **51.2** |
| | MATH500 | 68.4 | 60.0 | 68.8 | 61.8 | **74.4** | **73.0** |
| | GSM8K | 82.1 | 71.2 | 79.6 | 79.3 | **84.5** | **85.8** |
| | Average | 47.8 | 36.8 | 47.6 | 44.7 | **52.8**$_{/+5.0}$ | **53.2**$_{/+5.4}$ |
| 4B | AIME25 | 23.3 | 22.5 | 22.5 | | **30.4** | **28.3** |
| | OlympiadBench | 47.7 | 45.0 | 50.1 | | **50.5** | **52.0** |
| | AMC23 | 62.8 | 60.9 | 64.1 | OOM | **70.3** | **70.6** |
| | MATH500 | 82.8 | 76.1 | 83.2 | | **84.4** | **85.6** |
| | GSM8K | 90.5 | 85.3 | **90.9** | | 90.4 | **91.5** |
| | Average | 61.4 | 58.0 | 62.2 | – | **65.2**$_{/+3.8}$ | **65.6**$_{/+4.2}$ |

(iteration-decider training), the neural decider successfully imitates the oracle strategy, reaching about 83% accuracy at predicting iteration decisions of oracle labels, as shown in Figure 10.

**Adding Iteration Depth**. We train a 1.7B TaH with maximum three iteration (TaH-3). TaH-3 yields 5.8% average gain over Standard, and 0.8% over TaH-2. Detailed results are in Appendix A.2.4.

**Generalizability**. We further study generalization when TaH is evaluated out of domain (OOD) or trained on broader data mixtures. First, when trained only on math data, TaH+ still improves OOD STEM performance on MMLU-STEM (4.7% and 2.9% for 0.6B and 1.7B respectively), indicating that the learned thinking patterns transfer robustly across domains. Second, finetuning Qwen3-1.7B-Base on a balanced OpenR1 mixture of math, QA, and code shows that TaH+ yields consistent gains over Standard across all categories, improving the overall average accuracy by 6.8%. See additional experiment and performance details in Appendix A.2.1.

## 5.3 DESIGN CHOICE EXPLORATION

We demonstrate the effectiveness and robustness of TaH by finetuning our model with alternative model architectures and training schemes, or altering the continuation thresholds. All results are reported on MATH500, AMC23 and OlympiadBench (Olym.).

**Model Architecture**. (1) **Iteration Scheme**. As shown in Table 2, TaH's dynamic iteration scheme outperforms the *Standard* and *AlwaysThink* alternatives, confirming the benefit of avoiding latent underthinking and overthinking. Note that for *Standard*, duo-causal attention degenerates to regular causal attention. (2) **Duo-Causal Attention**. Replacing duo-causal attention with standard causal attention variants causes significant drops: (a) attending only to the first iteration (Causal-iter1) drops 5.4%; (b) attending only to the current iteration (Causal) drops even larger at 8.5%. The latter failure confirms duo-causal attention's essential role for cross-depth information flow. (3) **LoRA and residual connections**. Removing LoRA and residual connections leads to consistent drops, confirming their beneficial roles in objective transition across iterations.

Table 2: Ablation of iteration scheme, attention mechanism and architecture designs on TaH-0.6B.

| Ablation | Iter. Scheme | Attention | LoRA | Residual | MATH500 | AMC23 | Olym. | Average |
|---|---|---|---|---|---|---|---|---|
| Base | TaH | Duo-causal | ✓ | ✓ | **51.2** | **32.5** | **23.9** | **35.9**/+0.0 |
| Scheme | Standard | Duo-causal | ✓ | ✓ | 47.2 | 23.4 | 18.8 | 29.8/−6.1 |
|  | AlwaysThink |  |  |  | 32.8 | 15.6 | 10.2 | 19.5/−16.4 |
| Attention | TaH | Causal-iter1 | ✓ | ✓ | 47.8 | 24.4 | 19.4 | 30.5/−5.4 |
|  |  | Causal |  |  | 42.0 | 23.8 | 16.4 | 27.4/−8.5 |
| Arch. | TaH | Duo-causal | ✗ | ✓ | **51.6** | 29.7 | 22.4 | 34.6/−1.3 |
|  |  |  | ✗ | ✗ | 49.2 | 22.5 | 21.2 | 31.0/−4.9 |

Table 3: Ablation study on training schemes.

| Supervision | Iter. Policy | MATH500 | AMC23 | Olympiadbench | Average |
|---|---|---|---|---|---|
| Token-only | Oracle | **51.2** | **32.5** | **23.9** | **35.9** |
| *Token+latent* | Oracle | 49.4 | 29.6 | 15.9 | 31.6 /−4.3 |
| Token-only | *Iter. decider* | 44.8 | 24.1 | 17.3 | 28.7 /−7.2 |
|  | *Dynamic* | 11.0 | 2.8 | 2.7 | 5.5/−30.4 |

**Training Scheme**. (1) **Supervision type.** Inspired by early exit methods, a common alternative supervises all iteration depths with next-token labels to enable flexible early termination. It enforces accurate prediction at depth 1 even for hard tokens. As shown in Table 3, such *token+latent* supervision underperforms our token-only approach that supervises only at oracle-determined depths. It aligns with our intuition that different iterations should focus on tokens of different difficulties. (2) **Iteration policy during LLM training.** We compare our static oracle strategy $\pi$ with two alternatives. The *iter. decider* trains the decider first then uses it during backbone training, but suffers from the coupling challenge discussed in Section 4.3. The *dynamic* recalculates the oracle using the evolving backbone in Equation 7, encountering the same coupling challenge and causing training collapse. These results support our backbone training recipe: using next-token supervision with oracle iteration policy.

**Continuation Threshold**. As shown in Figure 4, TaH maintains robust performance across different continuation thresholds and iteration ratios. We empirically set $c_{\text{threshold}} = 0.9$ for all evaluations.

## 5.4 BEHAVIOR ANALYSIS

**Latent Overthinking**. To analyze latent thinking patterns, we verbalize tokens from all iteration depths using their last-layer hidden states. The oracle method uses the oracle policy $\pi$ from Section 4.3 for iteration decision. (1) **Generation.** Since ground-truth tokens are unavailable during generation, we use predictions from the stronger DeepSeek-R1-Distill-Qwen-32B model (Guo et al., 2025) as proxy labels. Table 4 shows that the oracle policy substantially improves performance by verbalizing correct predictions immediately while iterating only on incorrect ones. With our trained

| Training | Inference | Accuracy |
|---|---|---|
| Standard | Standard | 52 |
| AlwaysThink | AlwaysThink | 38/−14 |
| AlwaysThink | TaH-Oracle | 77/+25 |
| TaH-Oracle | TaH-Decider | 54/+ 2 |
| TaH-Oracle | TaH-Oracle | 80/+28 |

Table 4: Impact of iteration schemes on Qwen3-0.6B (first 100 MATH500 samples).

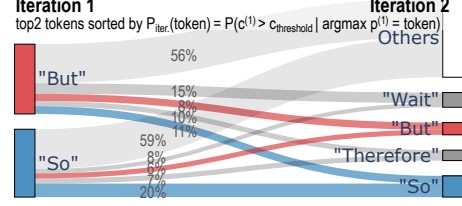

Figure 5: Next-token prediction changes across iterations. Top2 tokens that think-twice most are visualized.

iteration decider approximating the oracle, TaH outperforms both Standard and AlwaysThink baselines. However, the ideal oracle policy achieves even higher gains, indicating future potential. (2) **Next-token prediction**. We evaluate next-token prediction accuracy on the Open-R1 dataset, using the test model itself as the reference model in $\pi$. Figure 1 reveals that AlwaysThink produces more incorrect than correct revisions, demonstrating latent overthinking. In contrast, oracle-controlled iterations substantially increase correct revisions by selectively targeting hard tokens.

**Token Alternation Patterns**. We analyze which tokens TaH selects for deeper iteration. On the validation set, *But* and *So* emerge as top candidates, with iteration probabilities of 34% and 18%, respectively. These tokens signal critical contrasting or causal relationships, confirming that models benefit from additional processing at logically complex junctures. Figure 5 illustrates how TaH alternates predictions after iteration at these key tokens, suggesting logic refinement behavior. See Appendix A.3.4 for detailed analysis.

**Attention Pattern**. We visualize the attention pattern of TaH. As discussed in Figure 11 and Appendix A.3.6, the duo-causal attention automatically focuses on different iterations in different heads, extracting broader contexts from multiple depths.

**Iteration and FLOPs**. Tables 10 and 11 report the average iteration count, per-token FLOPs, and memory access cost of TaH. TaH matches the FLOPs of the Standard baseline (averaging 1.06 iterations per token), while significantly undercutting AlwaysThink (2.00 iterations), which incurs $\approx 2.2\times$ FLOPs and memory access. When tested on an NVIDIA A800-80GB GPU, TaH achieves a $2.48\times$ speedup over AlwaysThink and reduces peak memory usage by $1.48\times$. See Appendices A.2.2 and A.2.3 for more details.

# 6 CONCLUSION

We present TaH, a selective latent thinking method that iterates deeper only on hard tokens. Architecturally, TaH introduces duo-causal attention, depth-specific LoRA, and a neural iteration decider to facilitate dynamic depths. An oracle policy guides the stable two-stage training for the tightly coupled LLM backbone and decider. Across five reasoning benchmarks, TaH improves accuracy by 4.0-5.4% over strong baselines with minimal overhead ($<$3% additional parameters and $\approx$6% extra iterations), establishing a new paradigm for better reasoning within the current parameter budgets.

## ETHICS STATEMENT

This study raises no ethical issues, it did not involve human subjects or sensitive personal data.

## REPRODUCIBILITY STATEMENT

This paper provides sufficient information to reproduce the reported results. All experiments were conducted using publicly available datasets together with open-source models and code. Appendix A details implementation aspects, including data selection, hyperparameters, and training procedures. To facilitate full reproducibility, we will release the code, configuration files, and model checkpoints upon publication.

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

# A APPENDIX

## A.1 ADDITIONAL EXPERIMENT SETUPS

### A.1.1 TRAINING RECIPE

We follow the official training setup of Open-R1 (Hugging Face, 2025). For Standard, TaH, and TaH+, we use a maximum sequence length of 8192 tokens. For AlwaysThink, we reduce the maximum length to 4096 due to its substantially higher memory usage during training. Detailed training hyperparameters are listed in Table 5.

### A.1.2 BASELINE SETUPS

**Routing**. The query-level routing baseline selects a model from a candidate pair for each question. In our experiments, we use two pairs: (1) MobileLLM-R1-360M, Qwen3-1.7B, and (2) Qwen3-0.6B, Qwen3-4B. All candidate models are SFT-trained under the same settings as the Standard baseline (Section 5). For each pair, the routing ratio is calibrated so that the average active parameter count matches our 0.6B and 1.7B targets, respectively.

**AlwaysThink**. AlwaysThink uses the exact same architecture as TaH, but substitutes the iteration decider to one that always iterates twice.

### A.1.3 PARAMETER BREAKDOWN

Table 6 reports the parameter breakdown of the Standard, TaH, and TaH+ methods. To offset the additional parameters introduced by TaH through LoRA and the iteration decider, we remove one layer from the LLM backbone, ensuring a fair comparison. In practical deployments, we recommend TaH+, which adds only about 3% additional parameters.

### A.1.4 LATENT OVERTHINKING ANALYSIS SETUP

We set up an oracle experiment to estimate the performance upper bound of our method. The oracle employs the DeepSeek-R1-Distill-Qwen-32B model as a dynamic label generator, replacing the MLP-based iteration decider. During each iteration, we compare the token predictions from the label generator with those from the TaH model. The model continues to iterate only when the top-1 predictions of these two models differ. Due to resource constraints and computational overhead, we evaluated the accuracy only on the first 100 samples from the MATH500 dataset, denoted as MATH100 throughout the paper.

## A.2 ADDITIONAL EXPERIMENTAL RESULTS

### A.2.1 GENERALIZABILITY

**General Training and Evaluation**. To verify generalizability, we expanded our training and evaluation to diverse domains. We followed the exact protocol from the main paper to finetune Qwen3-1.7B-Base. The only modification was replacing the math-only dataset with a balanced subset of OpenR1 (100k samples total) covering all task splits, ensuring a fair comparison by maintaining the

Table 5: Training hyperparameters.

| Hyperparameter | Value |
| --- | --- |
| learning rate | 4e-5 |
| max grad norm | 0.2 |
| training epochs | 5 |
| global batch size | 128 |
| warmup ratio | 0.03 |
| lr scheduler | cosine (min-lr ratio 0.1) |
| precision | bfloat16 |

Table 6: Parameter breakdown of Standard, TaH, and TaH+. Counts are reported using M (million) and B (billion).

| Param. | Method | Backbone | LoRA | Iter. Decider | Total |
|--------|--------|----------|------|---------------|-------|
| *0.6B* | Standard | 596M | – | – | 596M |
|        | TaH | 580M | 10M | 5M | 595M |
|        | TaH+ | 596M | 10M | 5M | 611M |
| *1.7B* | Standard | 1.72B | – | – | 1.72B |
|        | TaH | 1.67B | 34M | 18M | 1.72B |
|        | TaH+ | 1.72B | 34M | 18M | 1.77B |

Table 7: Performance of Qwen3-1.7B models trained on a general OpenR1 mixture (math, QA, and code) across downstream benchmarks.

| Category | Dataset | Standard | SoftThink | AlwaysThink | TaH+ |
|----------|---------|----------|-----------|-------------|------|
| Math | MATH500 | 67.8 | 64.8 | 63.2 | **72.6** |
|      | AMC23 | 39.7 | 40.3 | 40.9 | **48.4** |
| QA | GPQA | 30.3 | 33.3 | 30.5 | **39.4** |
|    | MMLU-STEM | 74.1 | 73.5 | 69.6 | **76.6** |
| Code | HumanEval+ | 44.2 | 44.5 | 25.6 | **48.2** |
|      | MBPP+ | 27.2 | 27.8 | 16.4 | **39.0** |
| Average | | 47.2 | 47.4 | 41.0 | **54.0**/+6.8 |

same data scale. As shown in Table 7, TaH+ achieves consistent improvements across math, QA, and coding domains, with an average performance gain of 6.8%. This demonstrates TaH's effectiveness on diverse reasoning and generation tasks beyond pure mathematics.

**Out-Of-Domain (OOD) Performance**. We further evaluated the zero-shot generalization capability of models trained solely on math datasets from the main paper. As shown in Table 8, TaH+ demonstrates consistent improvements not only on in-domain math benchmarks (MATH500, AMC23) but also on out-of-domain tasks like MMLU-STEM. This indicates that the thinking patterns learned by TaH+ on math problems are robust and transferrable to broader scientific reasoning tasks.

### A.2.2 REAL-WORLD EFFICIENCY

Setup. We investigate the real-world efficiency of different 1.7B models under our current implementation. All measurements were obtained on a single A800 GPU with a batch size of 1 and a maximum output length of 8192 tokens, using a challenging AIME25 prob-

Table 8: Performance of math-only trained models (0.6B and 1.7B) on in-domain math benchmarks and the out-of-domain STEM benchmark (MMLU-STEM).

| Param. | Benchmark | Standard | SoftThink | AlwaysThink | TaH+ |
|--------|-----------|----------|-----------|-------------|------|
| *0.6* | MATH500 | 47.2 | 48.8 | 32.8 | **54.2** |
|       | AMC23 | 23.4 | 24.1 | 15.6 | **30.6** |
|       | MMLU-STEM | 51.6 | 51.4 | 42.6 | **56.3** |
|       | Average | 40.7 | 41.4 | 30.3 | **47.0** |
| *1.7* | MATH500 | 68.4 | 68.8 | 61.8 | **73.0** |
|       | AMC23 | 42.2 | 43.1 | 42.5 | **51.2** |
|       | MMLU-STEM | 70.8 | 70.6 | 63.8 | **73.7** |
|       | Average | 60.5 | 60.8 | 56.0 | **66.0** |

Table 9: Real-world decoding performance on a single A800 GPU, including maximum memory usage (GB), decoding latency (s), throughput (tokens/s), and per-component time breakdown.

| Metric | Standard | | TaH | | AlwaysThink | |
|---|---|---|---|---|---|---|
| Memory (GB) | 4.3 | | 4.6 | | 6.8 | |
| Latency (s) | 210.6 | | 301.4 | | 747.2 | |
| Throughput (tok/s) | 38.9 | | 27.2 | | 11.0 | |
| **Component** | **Latency (s)** | **Ratio(%)** | **Latency (s)** | **Ratio(%)** | **Latency (s)** | **Ratio(%)** |
| Iter-1 Forward | 210.6 | 100.0 | 229.8 | 76.2 | 224.1 | 30.0 |
| Iter-2 Forward | – | – | 29.6 | 9.8 | 384.7 | 51.5 |
| Iter. Decider | – | – | 10.5 | 3.5 | – | – |
| LoRA Switching | – | – | 7.5 | 2.5 | 91.1 | 12.2 |
| Other | – | – | 24.1 | 8.0 | 47.4 | 6.3 |

lem where all three methods reached the token limit. Memory usage was profiled using `torch.cuda.memory._record_memory_history`.

**Memory**. As shown in Table 9, TaH introduces minimal memory overhead of +7% over Standard, even at an extensive length of 8192 tokens. In contrast, AlwaysThink increases memory usage by 58%. This surge is primarily due to its dense iteration doubling the KV cache size, whereas TaH keeps the cache compact by skipping the second iteration for 94% of tokens.

**Latency Breakdown**. We report the decoding latency, throughput, and a detailed time breakdown for Standard, AlwaysThink, and TaH in Table 9. Here, *Iter-1 forward* and *Iter-2 forward* denote the total forward-pass time spent on the first and second latent iterations, respectively; *Iter decider* is the time for the iteration decider network to judge whether to continue iterating or verbalize; *LoRA switching* is the overhead of switching LoRA adapters; and *Other* includes tensor initialization, concatenation, and related bookkeeping.

**Discussion**. We note that our current implementation is not yet optimized at the system level, so there remains room for further efficiency improvements. For example, the *LoRA Switching* and *Other* overheads (bookkeeping) are relatively high due to the Python-level implementation of dynamic control flow. These engineering optimizations are largely orthogonal to the algorithmic design of TaH, and we plan to continue refining the implementation to further reduce latency and memory overhead. The theoretical FLOPs and memory access analysis of TaH are provided in Appendix A.2.3.

### A.2.3 THEORETICAL EFFICIENCY ANALYSIS

Following prior work Hoffmann et al. (2022); Yang et al. (2024); Ma et al. (2025), we analyze the computational and memory access overhead of TaH relative to the Standard and AlwaysThink baselines. Table 10 presents the average number of input/output tokens and latent iterations per token across five benchmarks. We use these statistics to calculate the theoretical computation and memory access costs for each method.

As shown in Table 11, TaH incurs only a marginal increase in cost per token (1.04 to 1.05×) compared to the Standard baseline. In comparison, *AlwaysThink* is prohibitively expensive, requiring 2.19 to 2.27× more computation and memory access. These theoretical results confirm that TaH exceeds the reasoning benefits of fixed-depth recurrent transformers without the substantial efficiency penalty.

### A.2.4 ITERATION DEPTH BEYOND TWO

**Hard Token Labeling**. Previous works have proposed many methods to evaluate the hardness of each tokens in the training data, like through excess loss (Lin et al., 2024; Xie et al., 2023), entropy (Wang et al., 2025b; Chen et al., 2023) and prediction difference (Fu et al., 2025a).

For shallow iteration budgets within two ($D_{\max} \leq 2$), we adopt the prediction difference policy. It simply labels the tokens that do not yield top-1 in next-token prediction at the first iteration as hard

Table 10: Input tokens (shared across methods) and output token / iteration statistics for Standard, AlwaysThink, TaH, and TaH+.

| Param. | Dataset | In. | Standard Out. | Iter. | AlwaysThink Out. | Iter. | TaH Out. | Iter. | TaH+ Out. | Iter. |
|---|---|---|---|---|---|---|---|---|---|---|
| 0.6B | AIME25 | 159 | 7450 | 1.00 | 7316 | 2.00 | 7648 | 1.05 | 7486 | 1.06 |
| | OlympiadBench | 100 | 6599 | 1.00 | 6622 | 2.00 | 6631 | 1.09 | 6513 | 1.06 |
| | AMC23 | 85 | 6377 | 1.00 | 6368 | 2.00 | 6242 | 1.05 | 6145 | 1.05 |
| | MATH500 | 71 | 4823 | 1.00 | 5350 | 2.00 | 4877 | 1.05 | 4793 | 1.06 |
| | GSM8K | 61 | 1955 | 1.00 | 2844 | 2.00 | 1923 | 1.07 | 1791 | 1.07 |
| | Average ratio | – | 1.00× | 1.00× | 1.02× | 2.00× | 1.00× | 1.06× | 0.97× | 1.06× |
| 1.7B | AIME25 | 159 | 7195 | 1.00 | 7173 | 2.00 | 7496 | 1.06 | 7498 | 1.06 |
| | OlympiadBench | 100 | 6008 | 1.00 | 6484 | 2.00 | 6387 | 1.06 | 6258 | 1.06 |
| | AMC23 | 85 | 5681 | 1.00 | 7543 | 2.00 | 6122 | 1.04 | 5852 | 1.06 |
| | MATH500 | 71 | 4004 | 1.00 | 4414 | 2.00 | 4233 | 1.06 | 4286 | 1.06 |
| | GSM8K | 61 | 1451 | 1.00 | 1644 | 2.00 | 1721 | 1.08 | 1686 | 1.08 |
| | Average ratio | – | 1.00× | 1.00× | 1.13× | 2.00× | 1.09× | 1.06× | 1.07× | 1.06× |

Table 11: Decoding computation (GFLOPs) and memory access (GB) per output token for Standard, AlwaysThink, TaH and TaH+ methods.

| Param. | Dataset | Standard Comp. | Mem. | AlwaysThink Comp. | Mem. | TaH Comp. | Mem. | TaH+ Comp. | Mem. |
|---|---|---|---|---|---|---|---|---|---|
| 0.6B | AIME25 | 1.47 | 1.38 | 3.35 | 3.14 | 1.52 | 1.43 | 1.57 | 1.47 |
| | OlympiadBench | 1.41 | 1.32 | 3.21 | 3.02 | 1.51 | 1.42 | 1.50 | 1.41 |
| | AMC23 | 1.40 | 1.31 | 3.17 | 2.97 | 1.43 | 1.34 | 1.46 | 1.37 |
| | MATH500 | 1.31 | 1.22 | 2.98 | 2.80 | 1.35 | 1.26 | 1.39 | 1.31 |
| | GSM8K | 1.14 | 1.06 | 2.54 | 2.38 | 1.19 | 1.12 | 1.22 | 1.14 |
| | Average ratio | 1.00× | 1.00× | 2.27× | 2.27× | 1.04× | 1.04× | 1.06× | 1.06× |
| 1.7B | AIME25 | 4.31 | 4.03 | 9.45 | 8.83 | 4.51 | 4.21 | 4.64 | 4.34 |
| | OlympiadBench | 4.16 | 3.88 | 9.18 | 8.58 | 4.36 | 4.07 | 4.48 | 4.18 |
| | AMC23 | 4.12 | 3.85 | 9.54 | 8.91 | 4.24 | 3.96 | 4.43 | 4.13 |
| | MATH500 | 3.92 | 3.66 | 8.45 | 7.89 | 4.10 | 3.83 | 4.23 | 3.95 |
| | GSM8K | 3.62 | 3.38 | 7.48 | 6.98 | 3.87 | 3.61 | 3.98 | 3.72 |
| | Average ratio | 1.00× | 1.00× | 2.19× | 2.19× | 1.05× | 1.05× | 1.08× | 1.08× |

tokens. Formally, we use a binary halting rule:

$$H_i^\pi = \begin{cases} 0, & \text{if } h_i = 0 \quad \text{(easy token)} \\ D_{\max}, & \text{if } h_i = 1 \quad \text{(hard token)} \end{cases} \quad (9)$$

If the iteration depth goes beyond 2 ($D_{\max} > 2$), we use the reference model's cross-entropy as a non-binary indicator of difficulty. Define

$$\ell_i^{\text{ref}} = -\log p_{i,\text{ref}}^{(0)}(t_{i+1}).$$

We then map difficulty to halting depth via monotone quantile binning:

$$H_i^\pi = \left\lfloor \text{QuantileRank}(\ell_i^{\text{ref}}) \cdot D_{\max} \right\rfloor, \quad (10)$$

where $\text{QuantileRank}(\cdot) \in [0, 1]$ is the empirical CDF over the training set (higher loss $\Rightarrow$ deeper halting). This induces per-depth continuation labels $c_i^{(d)} = \Vmathbb{1}[d < H_i^\pi]$ for $d \in \{0, 1, \ldots, D_{\max}\}$.

**Experiment Result**. Specifically, we train a 1.7B TaH with a maximum per-token iteration count of 3, using oracle labels generated by the method described above. As shown in Table 12, TaH-3 achieves a further improvement of **0.8%** on average over TaH-2.

Table 12: Performance comparison between TaH-2 and TaH-3 (maximum per-token iterations of 2 and 3, respectively). Iter.2 and Iter.3 denote the per-token percentages executing two and three iterations, respectively.

| Param. | Dataset | Standard | TaH-2 | | TaH-3 | | |
|---|---|---|---|---|---|---|---|
| | | Acc. | Acc. | Iter.2 | Acc. | Iter.2 | Iter.3 |
| | MATH500 | 68.4 | **74.4** | 5.6 | 72.6 | 5.3 | 0.2 |
| | GSM8K | 82.1 | 84.5 | 7.5 | **84.8** | 7.6 | 0.3 |
| *1.7B* | AMC23 | 42.2 | 48.4 | 4.2 | **48.7** | 5.1 | 0.1 |
| | OlympiadBench | 33.0 | 38.8 | 5.7 | **41.6** | 5.4 | 0.2 |
| | AIME25 | 13.3 | 17.9 | 6.0 | **20.4** | 5.3 | 0.1 |
| | Average | 47.8 | 52.8 | 5.8 | **53.6** | 5.7 | 0.2 |

Table 13: Performance on MATH500 and GSM8K-500 (first 500 GSM8K samples)

| Dataset | Method | |
|---|---|---|
| | Standard-0.6B | Ponder-1.4B |
| MATH500 | 47.2 | 2.0 |
| GSM8K-500 | 62.8 | 1.8 |
| Avg. | 55 | 1.9 |

### A.2.5 ADDITIONAL LATENT THINKING METHODS

Some latent thinking methods requires pre-training and uses base model other than Qwen3. We also compare with these methods, including Ponder (Zeng et al., 2025). Specifically, we adopt the released pretrained PonderingPythia-1.4B as the base model and perform SFT on the same training data. We observe that the fine-tuned model learns the stylistic patterns of the training data, but still underperforms substantially, which may be attributable to the limited capability of the PonderingPythia-1.4B backbone.

### A.2.6 TRAINING RECIPE INFLUENCE

Figure 13 expands Table 3 by showing validation perplexity dynamics across different supervision signals and iteration policies. The naming convention matches Table 3. TaH with token-only supervision and the oracle policy yields lower perplexity than *iter. decider* and *token+latent*. Although the *dynamic* policy achieves the lowest perplexity, it fails on downstream tasks and often produces infinite-loop generations.

## A.3 ADDITIONAL ANALYSIS

### A.3.1 ORACLE POLICY AND HARD TOKEN ANALYSIS

**Metrics for Hard Token Labeling**. We investigate different metrics for defining hard tokens to validate our choice of top-1 prediction mismatch. We compare three labeling strategies:

1. Top-1 Mismatch (TaH Default): Labels a token as hard if the reference model's greedy prediction differs from the ground truth.

2. Entropy (TaH-Entropy): Labels a token as hard if the reference model's prediction entropy exceeds a threshold.

3. Cross-Entropy (TaH-CE): Labels a token as hard if the reference model's cross-entropy loss exceeds a threshold.

To ensure a fair comparison, for *TaH-Entropy* and *TaH-CE*, we set the thresholds such that the number of hard tokens in each sample matches the total ratio from the default Top-1 Mismatch policy. This isolates the impact of *which* tokens are selected, rather than *how many*.

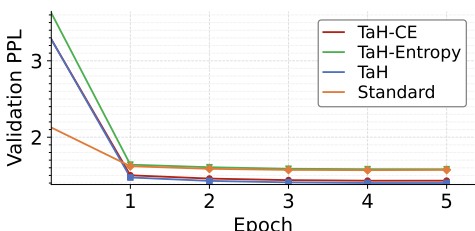

Figure 6: Validation loss curves of 0.6B models trained with different oracle labeling metrics. The default Top-1 Mismatch policy yields the lowest validation loss.

Table 14: Performance comparison of different difficulty metrics (Entropy, Cross-Entropy, and Top-1 Accuracy) on 0.6B models. All methods mark the same total number of tokens as "hard."

| Method | MATH500 | AMC23 | OlympiadBench | Average |
|---|---|---|---|---|
| TaH-Entropy | 42.0 | 21.9 | 16.9 | 26.9 |
| TaH-CE | 47.4 | 21.2 | 20.4 | 29.7 |
| TaH | **51.2** | **32.5** | **23.9** | **35.9** |

Figure 6 compares the validation loss, and Table 14 reports downstream accuracy on 0.6B models. While cross-entropy (*TaH-CE*) improves over entropy labeling (*TaH-Entropy*), the Top-1 Mismatch policy (*TaH*) achieves superior performance across all benchmarks. This empirically verifies that directly targeting tokens where the model's top-1 prediction is wrong is the most effective way to identify hard tokens for TaH.

**Labeling Robustness**. We investigate the robustness of hard-token labels with respect to the choice of reference model. We do so by analyzing the consistency of hard-token identification across different model scales (e.g., Qwen3-0.6B, 1.7B, and 4B).

First, we quantify the agreement between models. As shown in Figure 7, hard tokens exhibit high consistency across scales. Notably, even a smaller, less accurate reference model (1.7B) successfully identifies 81% of the hard tokens for a larger model (4B).

Second, to understand the quality of this overlap, we partition tokens into an *overlap set* (marked as hard by both models) and a *non-overlap set* (marked as hard by only one model). We plot the cross-entropy loss under each reference model in Figure 8. We observe that overlap tokens have substantially higher average cross-entropy ($\approx 2.0\times$) than non-overlap tokens for *all* reference models. This indicates that either reference model can identify this core set of "hard" tokens, which corresponds to positions of genuine, high uncertainty. It reveals a consensus on hardness among models even of different sizes.

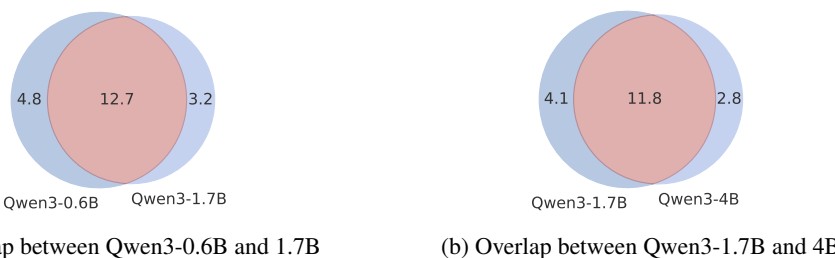

(a) Overlap between Qwen3-0.6B and 1.7B          (b) Overlap between Qwen3-1.7B and 4B

Figure 7: Venn diagrams illustrating the overlap of hard-token labels between different reference models. The high overlap proportions indicate that "hard" tokens are largely consistent across model scales.

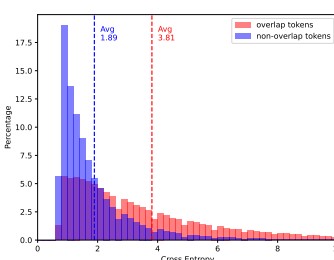

(a) Qwen3-1.7B: cross-entropy of overlap vs. non-overlap hard tokens (w.r.t. Qwen3-0.6B).

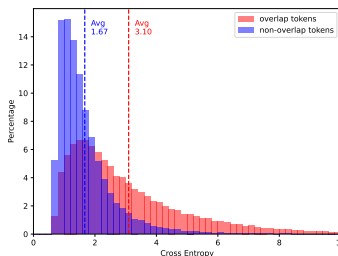

(b) Qwen3-0.6B: cross-entropy of overlap vs. non-overlap hard tokens (w.r.t. Qwen3-1.7B).

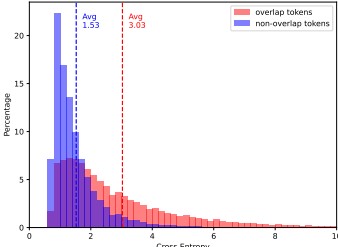

(c) Qwen3-4B: cross-entropy of overlap vs. non-overlap hard tokens (w.r.t. Qwen3-1.7B).

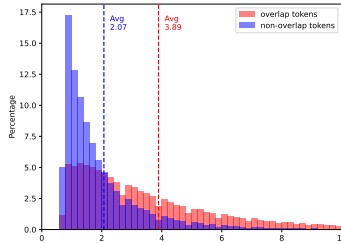

(d) Qwen3-1.7B: cross-entropy of overlap vs. non-overlap hard tokens (w.r.t. Qwen3-4B).

Figure 8: Token-level cross-entropy distributions of overlap and non-overlap hard tokens across different reference model pairs. For each pair of reference models (e.g., Qwen3-1.7B and Qwen3-0.6B), we plot the cross-entropy of tokens labeled as hard by both models (overlap) and by only one model (non-overlap) on both reference models.

Table 15: Iteration decider behavior and downstream gains on different validation subsets. The decider is trained once on general OpenR1 and evaluated without retraining.

| Metric | Math | Code | QA |
|---|---|---|---|
| Iteration Percentage | 7.8% | 10.7% | 26.6% |
| Iteration Accuracy | 86.7% | 82.3% | 76.6% |
| Benchmark Gain over Standard | **+6.8%** | **+7.9%** | **+5.8%** |

### A.3.2 ITERATION DECIDER ROBUSTNESS

We evaluate the iteration decider, trained on the general OpenR1 corpus, across three validation subsets (Math, Code, and QA) to quantify its robustness and cross-domain generalizability. As summarized in Table 15, the decider maintains high decision accuracy across all domains without any retraining.

Despite being invoked on only 7.8-26.6% of tokens, the decider consistently yields 5.8-7.9% absolute accuracy gains over the standard single-pass baseline on all three domains. Moreover, the decider automatically adjusts its iteration rate according to task difficulty: it iterates more frequently on QA (26.6%) than on Math (7.8%), even under a fixed threshold $c_{threshold} = 0.9$. This behavior indicates that the decider responds to intrinsic uncertainty signals in the model's predictive distribution rather than memorizing domain-specific patterns, consistent with the token-level difficulty analysis in Appendix A.3.3.

### A.3.3 HARD TOKEN IDENTIFIABILITY

Why is the iteration decider robust and generalizable across tasks? We investigate this by analyzing the intrinsic properties of "hard" tokens.

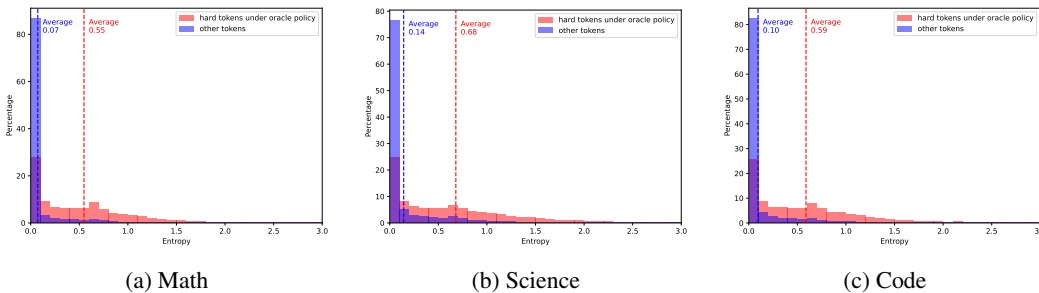

|(a) Math|(b) Science|(c) Code|

Figure 9: Output Logit entropy distribution at the first iteration of TaH, categorized by oracle policy's difficulty labels (hard token) on the OpenR1 validation set (Math, QA, Code). The distinct separation between distributions confirms that TaH's internal logits provide a strong, task-agnostic signal for identifying hard tokens.

Table 16: Conditional probabilities of continuation confidence and next-token distribution.

| Token $T_1$ | $P(c^{(1)} > c_{\text{threshold}} \mid t^{(1)} = T_1)$ | Token $T_2$ | $P(t^{(2)} = T_2 \mid t^{(1)} = T_1)$ |
|---|---|---|---|
| But | 34.3% | So | 13.63% |
| | | Wait | 12.17% |
| | | Therefore | 8.95% |
| So | 17.7% | So | 28.17% |
| | | Therefore | 13.67% |
| | | But | 4.89% |

We compute the token entropy of hard and easy tokens across three diverse subsets of the OpenR1 dataset (Math, Science, and Code). As shown in Figure 9, hard tokens exhibit a universal signature of significantly higher entropy ($> 5\times$) compared to easy tokens. This distinct separation confirms that "hardness" is an intrinsic, robustly identifiable property of the model's predictive state, rather than a complex, task-specific pattern. Given this clear signal, the neural iteration decider can easily learn reliable classification strategies that generalize well across different domains.

### A.3.4 TOKEN ALTERNATION PATTERN

We analyze tokens that most frequently trigger a second iteration ("think-twice" tokens). For each token type $t$, we compute the continuation rate

$$\Pr\left(c_i^{(1)} > c_{\text{threshold}} \mid t_i = t\right),$$

using the inference threshold $c_{\text{threshold}} = 0.9$ (Section 4.3). We estimate this quantity on the Open-R1 validation set and, for diagnostics, randomly sample 10K token positions ($\approx 0.4\%$ of tokens) to track whether the next-token prediction switches between depth 1 and depth 2. This setting quantifies which token types most often trigger an additional iteration and how often iteration alters the predicted next token.

### A.3.5 ITERATION DECISION ERROR

We analyze how iteration decision accuracy affects TaH's end-to-end response quality, since iteration decider will not be perfect as shown in Figure 10. To this end, we randomly inject errors into the oracle iteration-decider predictions at different rates. Formally, we denote the original oracle prediction as the *label* $l \in \{0, 1\}$ and the altered prediction as the *output* $o \in \{0, 1\}$. We define the *iter. error* as the total proportion of deliberately introduced errors:

$$\text{iter. error} = P(l \neq o) = \underbrace{P(l=1, o=0)}_{\text{underthink rate}} + \underbrace{P(l=0, o=1)}_{\text{overthink rate}}. \tag{11}$$

We further distinguish the impacts of overthinking and underthinking. Here, overthinking refers to cases where the decider incorrectly signals *continue*, while underthinking corresponds to cases

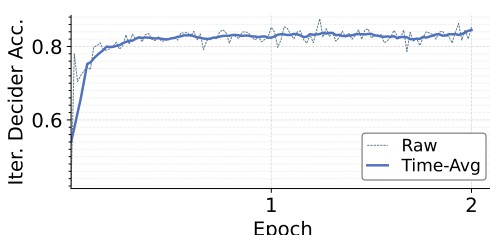

Figure 10: Iteration-decider accuracy vs. epoch (Qwen3-0.6B).

Table 17: TaH performance under different iteration-decider error rates. All values are reported in percentages.

| Iter. Error (%) | Underthink (%) | Overthink (%) | MATH100 Accuracy (%) |
|---|---|---|---|
| 0.0 | 0.0 | 0.0 | 80.0 |
| 2.8 | 2.8 | 0.0 | 78.0 |
| 10.0 | 1.5 | 8.5 | 55.4 |
| 15.0 | 2.1 | 12.9 | 45.2 |
| 20.0 | 2.5 | 17.5 | 27.1 |
| 22.1 | 0.0 | 22.1 | 21.6 |

where it incorrectly signals *stop*. Table 17 shows how TaH's MATH100 accuracy varies with different iteration error rates. We quantify these effects by fitting a linear model to the data:

$$\text{accuracy} = -1.41 \times \text{underthink rate} - 2.73 \times \text{overthink rate} + 0.81.$$

This analysis indicates that inaccurate iteration decisions are the main factor behind the performance gap between TaH and its oracle variant, with overthinking being the dominant source of performance gaps.

### A.3.6 DUO-CAUSAL ATTENTION PATTERN

We perform forward computation on 100 samples, each with a length of 128 tokens. Figure 11 shows the average attention weights of three representative attention heads in the second iteration of the TaH model. The left panel illustrates a head that mainly attends to keys from the first iteration. The middle panel shows a head focusing on keys from the second iteration. The right panel displays a head with a balanced attention distribution. These results suggest that the TaH model, under the duo-causal attention mechanism, can automatically learn diverse attention patterns across layers and heads.

Figure 12 further presents the total attention scores assigned to keys in the first iteration. It can be seen that the first layer tends to focus more on keys from the second iteration. Different layers also exhibit varying attention behaviors.

### A.4 IMPLEMENTATION DETAILS

### A.4.1 DUO-CAUSAL ATTENTION IMPLEMENTATION

Figure 14 illustrates the implementation of duo-causal attention, with the formal definitions provided below.

**(1) KV cache concatenation.** At depth $d$, we form the visible K/V sequence by concatenating all shallower-to-current depths along the sequence dimension:

$$\text{KV}^{(\leq d)} = [\,\text{KV}^{(1)}\,;\,\text{KV}^{(2)}\,;\,\cdots\,;\,\text{KV}^{(d)}\,].$$

This realizes the accessible set in Equation 5, allowing deeper iterations to access all shallower iterations while preserving positional causality. The KV cache is managed by iteration depth during

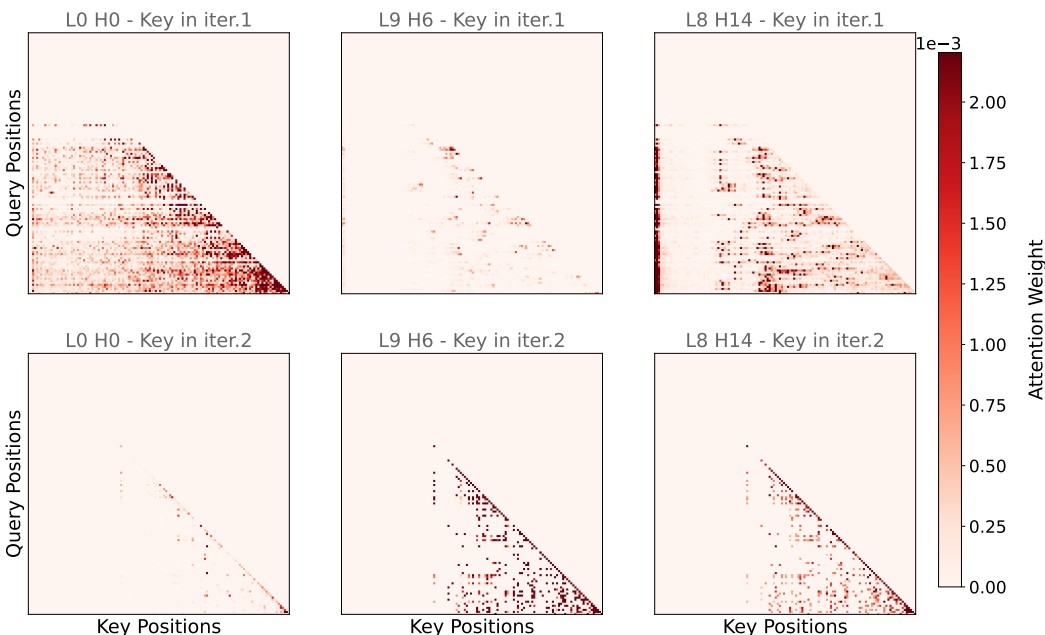

Figure 11: TaH duo-causal attention pattern.

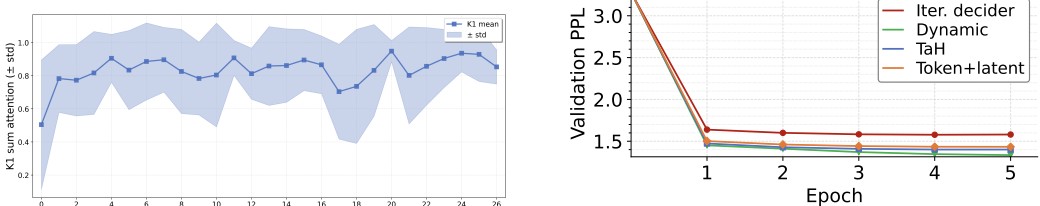

Figure 12: TaH mean and standard deviation of attention weights (key from iteration 1) across layers in iteration 2.

Figure 13: Validation perplexity for different training schemes.

decoding, as shown in Figure 14(b). The fragmented KV-cache management strategy is standard in existing LLM serving systems (Kwon et al., 2023; Zheng et al., 2024).

**(2) Two-dimensional causal mask.** For a query $(i, d)$, a key $(j, k)$ is attendable iff $j \leq i$ and $k \leq d$. We implement this as an additive attention mask with 0 for allowed entries and $-\infty$ otherwise, enforcing positional and iteration causality jointly. Figure 14(c) visualize the landscape of the duo-causal attention mask. When $d = 1$ for all tokens, the rule reduces to standard causal attention.

**(3) Compatibility with efficient attention.** The mask is provided in the standard additive form and the concatenated K/V remain contiguous along the sequence dimension, matching the usual scaled dot-product attention interface. As a result, duo-causal attention is directly compatible with optimized kernels such as FlashAttention, without kernel modifications.

## A.5 ADDTIONAL RELATED WORK

Instead of using the shared model parameter multiple times through latent iteration, previous work also proposes layer skipping methods for dynamic computing allocation.

**Layer Skipping**. Layer skipping aims to accelerate LLM inference by dynamically bypassing certain layers for specific tokens. Some methods use a learnable module to make real-time skipping decisions. MoD (Raposo et al., 2024) uses a top-k router to select a subset of tokens for processing,

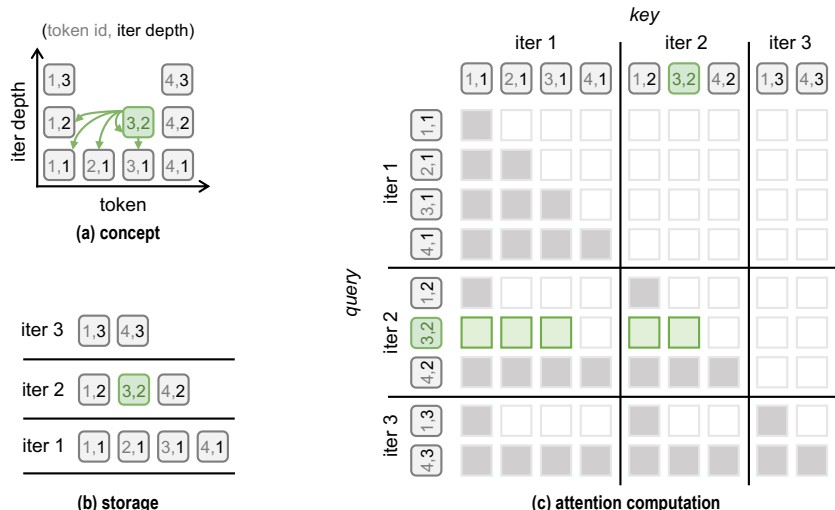

Figure 14: Duo-causal attention implementation. (a) Conceptual TaH example with dynamic iteration depths. Each cell denotes a token–depth pair (token id, iter depth). (b) Each iteration maintains its own KV cache. (c) KV caches from all iterations are concatenated into a 1D sequence and processed with standard attention under a duo-causal mask. The duo-causal mask is conceptually partitioned into blocks by iteration depth. The diagonal blocks use a standard causal mask, while off-diagonal blocks use reduced causal masks that enforce the duo-causal rules.

while FlexiDepth (Luo et al., 2025) uses a plug-in router to determine whether a layer should be bypassed. Others use a fixed strategy to skip layers. SkipDecode (Del Corro et al., 2023) enforces a monotonically decreasing number of active layers during generation. However, these methods still require loading the entire model's parameters, resulting in a large memory access overhead. Instead of skipping some layers, TaH adds computational depth by allowing core tokens to undergo multiple refinement iterations. This approach provides greater computational depth without increasing the model's parameter count.

### A.6 LIMITATIONS AND FUTURE WORK

**Comparison with Official Qwen3 Models**. Official Qwen3 models are trained on different data distributions and scales, and use different training procedures, including on-policy distillation (Yang et al., 2025). By contrast, our models use SFT only on limited, publicly accessible data. Consequently, performance may differ between the two.

**Future Work**. We consider advanced training techniques largely orthogonal to TaH. Promising directions include: (1) integrating TaH with online distillation to provide stronger supervision; (2) applying reinforcement learning to optimize the compute–accuracy trade-off and to learn continuation policies beyond imitation. Due to resource constraints, this work focuses on SFT with public data; we leave these combinations for future work.

### A.7 USE OF LARGE LANGUAGE MODELS (LLMS)

Large language models (e.g., GPT-5, Gemini) were used exclusively for proofreading and minor copyediting. They were not employed to generate scientific content, contribute to research ideation, or design methods. All research ideas, methods, and conclusions are solely the responsibility of the authors.

### A.8 EXAMPLES

We use an example question from AIME25 and responses from Standard, TaH and AlwaysThink to provide an intuitive understanding of our method. Only TaH can successfully answer this question. Blue-highlighted tokens indicate positions that undergo two iterations.

---

**Text 1. Question**

On $\triangle ABC$ points $A, D, E$, and $B$ lie that order on side $\overline{AB}$ with $AD = 4$, $DE = 16$, and $EB = 8$. Points $A, F, G$, and $C$ lie in that order on side $\overline{AC}$ with $AF = 13$, $FG = 52$, and $GC = 26$. Let $M$ be the reflection of $D$ through $F$, and let $N$ be the reflection of $G$ through $E$. Quadrilateral $DEGF$ has area 288. Find the area of heptagon $AFNBCEM$.

---

## Text 2. Standard Response

Okay, let me try to figure out this problem step by step. It's about a heptagon formed by reflecting points and finding its area. ... We have triangle $ABC$. On side $AB$, points $A, D, E$, and $B$ are in that order. The lengths are given as $AD = 4$, $DE = 16$, and $EB = 8$. So $AB = 28$ ... On side $AC$, points $A, F, G$, and $C$ are in that order with $AF = 13$, $FG = 52$, and $GC = 26$. So $AC = 91$ ... Let me place point $A$ at $(0,0)$ and $B$ at $(28,0)$ ... Let $C$ be at $(91, h)$ ... Then $D = (4,0)$, $E = (20,0)$, $F = (13, \frac{13h}{91})$, $G = (65, \frac{65h}{91})$ ...

Area of quadrilateral $DEGF$ can be calculated ... Applying the shoelace formula ... Area $= \frac{1}{2} \cdot \frac{1248h}{91} = \frac{624h}{91} = 6h$

So with 288: $6h = 288 \Rightarrow h = \frac{288}{6} = 48$

Reflections: $M$ is the reflection of $D$ through $F$ and $N$ is the reflection of $G$ through $E$ ... $M = (22,0)$, $N = (-39,0)$, $B = (28,0)$, $C = (91,48)$, $E = (20,0)$ ...

Using the shoelace formula for $AFNBCEM$ in order ... Sum $= 0 + 0 + 0 + 1344 + (-960) + 0 + 0 = 384$

Take absolute value and multiply by 1/2: Area $= \frac{1}{2} \cdot |384| = 192$

...

**Final Answer** $\boxed{192}$ ✗

## Text 3. TaH Response

Okay, let's try to tackle this geometry problem. It seems a bit complex with all these points and reflections, ... First, let me visualize triangle ABC. Points D, E, and B are on side AB with AD = 4, DE = 16, and EB = 8. ... Let me set up a coordinate system. Let's place point A at the origin $(0,0)$ ... Then, since $AB$ is a line segment, let me assume $AB$ is along the $x$-axis. So point $B$ would be at $(28,0)$ ... Now, points $F, G, C$ are on $AC$ ... Let me define coordinates for $C$ as $(c_x, c_y)$ ...

...

Compute the sum of $x_i y_{i+1} - x_{i+1} y_i$ for each $i$ ...

Sum of all terms: $0 + \frac{1300 c_y}{91} + 0 - > \frac{52 c_y}{91} = \frac{(1300-52)c_y}{91} = \frac{1248 c_y}{91}$

Then, the area is $\frac{1}{2} \cdot \left| \frac{1248 c_y}{91} \right| = \frac{624 c_y}{91}$

...

Compute the area of this heptagon ...

Using shoelace formula

...

Term2: –240  Term3: 840  Term4: 1176  Term5: –840  Term6: 240 ...

Sum: $0 - 240 + 840 + 1176 - 840 + 240 + 0 = 1176$

Take absolute value and multiply by 1/2: Area $= \frac{1}{2} \cdot |1176| = 588$

...

**Final Answer** $\boxed{588}$ ✓

## Text 4. AlwaysThink Response

Okay, so I need to find the area of the heptagon AFNBCEM. Let me first try to visualize the problem. There's triangle ABC. On side AB, there are points A, D, E, and B. The order is A, D, E, B. The distances given are AD=4, DE=16, and EB=8. So, starting at A, moving 4 units to D, then 16 units to E, then 8 units to B. So, the total length of AB is AD + DE + EB = 4 + 16 + 8 = 28 units. Got that.

...

Points in order: D(4,0), E(20,0), F(13,h), G(65,h), back to D(4,0). Compute the terms:

...

Sum these terms: $0 + 20h - 52h - 4h = -36h$

Take absolute value and multiply by 1/2: Area $= (1/2) * |-36h| = (1/2)*36|h| = 18|h|$

But the area is given as 288, so $18|h| = 288 \Rightarrow |h| = 16$.

...

Now, apply the shoelace formula ... Therefore, total sum is 8640/17

Take absolute value and multiply by 1/2: Area $= (1/2) * |8640/17| = 8640/34 = 4320/17 \approx 254.1176$

...

Therefore, the answer is 4320/17.

**Final Answer** $\boxed{\dfrac{4320}{117}}$ ✗

