# OpenReview forum: "Think-at-Hard: Selective Latent Iterations to Improve Reasoning Language Models"
_ICLR.cc/2026/Conference — Submitted to ICLR 2026_

### Official Review · Reviewer_oeNk · 2025-10-21

**Soundness:** 2
**Presentation:** 2
**Contribution:** 3
**Rating:** 4
**Confidence:** 4

**Summary:**

This paper proposes a model architecture and training method that improve upon prior recurrent Transformer designs. It first identifies a key limitation of uniform latent iteration—processing all tokens indiscriminately—which can degrade already-correct predictions. To address this, it introduces a learned router that selectively re-encodes only difficult tokens, reducing redundant computation and preventing overthinking. In addition, the paper presents duo-causal attention, enabling tokens to attend not only to earlier positions but also to representations from shallower iteration depths, promoting cross-depth information flow.

The training is conducted in two stages: first, the model is trained under an oracle policy that uses ground-truth token difficulty (from an oracle language model) to supervise selective iteration; then, a lightweight decider network is trained to imitate this oracle routing policy. This staged approach stabilizes optimization and ensures that the backbone learns useful latent refinements before the router is introduced.

Experiments on multiple reasoning benchmarks (GSM8K, MATH500, AMC23, AIME25, OlympiadBench) demonstrate consistent gains over standard and uniform recurrent baselines, with only ~6% of tokens undergoing extra iterations. Ablation studies show that LoRA for subsequent iteration models, residues and duo-attention all contribute to the final performance.

**Strengths:**

- The paper presents a clear and well-motivated improvement to recurrent Transformer models by introducing selective latent iteration through a learned router, addressing the issue of “latent overthinking” in prior methods.
- The proposed two-stage training scheme (oracle-guided backbone training followed by decider imitation) is well designed and empirically effective, leading to stable optimization and efficient use of computation.
- Experimental results are comprehensive across multiple reasoning benchmarks, showing consistent gains with minimal extra compute and parameter overhead.

**Weaknesses:**

To tackle the overthinking by token selectivity, the paper introduces: (i) duo-causal attention; (ii) router and a corresponding training paradigm.

1. The two designs are largely orthogonal — duo-causal attention can function independently without token eviction and itself induces a form of soft selectivity through attention scores, while the router performs hard selection. It can be seen as a way to achieve sparsity and computational efficiency for duo-casual attention. Given that both introduce additional compute, memory, and operational complexity, their relationship and potential redundancy should be analyzed more thoroughly.

2. The rationale for adopting duo-causal attention, rather than retaining the simpler scheme of attending only to previous tokens within the same depth as in prior recurrent Transformers, is not well justified. Since subsequent iterations already employ LoRA, the existing attention formulation could potentially suffice without introducing a new structure.

The ablation study only contrasts full duo-causal attention with a degenerate variant that attends solely to the first layer, omitting intermediate alternatives. The reported results show that routing alone achieves performance comparable to soft-thinking, whereas duo-attention brings most of the additional gains. This leaves unclear what the standalone contribution of duo-causal attention is and whether its benefits justify the added complexity.

Besides, it is beneficial to include a brief explanation on how the 2d tokens are flatten into 1d order and how positional encoding is applied in the main text.

**Questions:**

1. How are these two methods related given that duo-attention itself provides a form of soft selectivity? What is the standalone contribution of duo-causal attention without the router?
2. What is the specific rationale for introducing duo-causal attention instead of retaining the simpler causal attention over previous tokens within the same depth, as in prior recurrent Transformers? Given that later iterations use LoRA adaptation, could similar improvements be achieved without adding duo-attention?

---

> ### Author Response · Authors · 2025-11-22
>
> We sincerely thank Reviewer oeNk for the clear feedback.
> We appreciate your valuable recognition of our **motivation, method design, training scheme, and comprehensive experiments**. We address the concerns below.
>
> ### W1, Q1: Relation between duo-causal attention and iteration decider. Is any of these redundant?
>
> > W1: The two designs are largely orthogonal — duo-causal attention can function independently without token eviction and itself induces a form of soft selectivity through attention scores, while the router performs hard selection. It can be seen as a way to achieve sparsity and computational efficiency for duo-casual attention. Given that both introduce additional compute, memory, and operational complexity, their relationship and potential redundancy should be analyzed more thoroughly.
> >
> > Q1: How are these two methods related given that duo-attention itself provides a form of soft selectivity? What is the standalone contribution of duo-causal attention without the router?
>
> We appreciate the reviewer's unified perspective.
> While both components contribute to "selection," they serve fundamentally different and complementary roles: the decider uniquely provides efficiency (hard selection), while duo-causal attention ensures accuracy under dynamic depths (information flow).
>
> 1. Iteration decider:
>     * **Efficiency:** The decider skips the second iteration for ~94% of tokens, reducing FLOPs by >2x over fixed-depth counterparts, which is only possible with hard selection.
>
>     * **Quality:** Like overthinking in explicit CoT, "more thinking" (even if weighted by soft attention) can hurt performance in easy scenarios. Iteration decider alleviates latent overthinking for better response quality.
>
> 2. Duo-causal attention:
>     * **Handling misaligned depths:** Once dynamic iteration creates a ragged structure (Figure 2(b), where tokens have different depths), standard attention fails.
>     For example, if token $t_i$ stops at depth 1 but $t_{i+1}$ continues to depth 2, standard attention at depth 2 loses access to $t_i$ (which is absent at depth 2). Duo-causal attention bridges this gap, enabling cross-depth information flow with full sequence parallelism.
>
> Thus, they are not redundant. The decider creates the efficient dynamic depths that duo-causal attention enables to function correctly.
> We have expanded the ablation study (Table 2), showing that altering either component degrades TaH performance by 6.1% and 5.4%, respectively.
>
> ### W2, Q2: Necessity of duo-causal attention
>
> > W2: The rationale for adopting duo-causal attention, rather than retaining the simpler scheme of attending only to previous tokens within the same depth as in prior recurrent Transformers, is not well justified. Since subsequent iterations already employ LoRA, the existing attention formulation could potentially suffice without introducing a new structure.
> The ablation study only contrasts full duo-causal attention with a degenerate variant that attends solely to the first layer, omitting intermediate alternatives.
> >
> > Q2: What is the specific rationale for introducing duo-causal attention instead of retaining the simpler causal attention over previous tokens within the same depth, as in prior recurrent Transformers? Given that later iterations use LoRA adaptation, could similar improvements be achieved without adding duo-attention?
>
> **1. Rationale.**
> LoRA and Duo-Causal Attention solve different problems. LoRA shifts the model's prediction objective, but it cannot fix missing context.
> Standard causal attention relies on all tokens to exist at the current depth. While it works for fixed-depth models, in TaH, ~94% tokens don't exist at depth 2.  Consequently, standard attention sees an incomplete history. Duo-causal attention fixes this by allowing depth 2 to "look back" at depth 1.
>
> **2. Empirical Results.**
> We add the requested ablation in Table 2: replacing duo-causal attention with standard causal attention constrained to the *current iteration depth* while keeping LoRA.
>
> |Iter. Policy|Attention|LoRA|MATH500|AMC23|Olym.|Average|
> |:--|:--|:--|--:|--:|--:|--:|
> |Iter. Decider|Duo-causal|✓|**51.2**|**32.5**|**23.9**|**35.9** (+0.0)|
> |Iter. Decider|Causal (iter 1 only)|✓|47.8|24.4|19.4|30.5 (−5.4)|
> |Iter. Decider|Causal (current iter)|✓|42.0|23.8|16.4|27.4 (−8.5)|
>
> The performance drop (−8.5%) confirms that duo-causal attention is crucial for dynamic iteration.

---

> ### Author Response · Authors · 2025-11-22
>
> ### W3: Routing baseline
>
> > The reported results show that routing alone achieves performance comparable to soft-thinking, whereas duo-attention brings most of the additional gains. This leaves unclear what the standalone contribution of duo-causal attention is and whether its benefits justify the added complexity.
>
> We clarify that "Routing" in our paper refers to *query-level routing* (switching models per question), not our internal iteration decider (see Section 5.1, Appendix A.1.2).
> We revised and double-specified this.
> We also enriched Table 2 for a clearer ablation study of all TaH components.
>
> ### W4: How 2D tokens flatten into 1D
>
> > Besides, it is beneficial to include a brief explanation on how the 2d tokens are flatten into 1d order and how positional encoding is applied in the main text.
>
> We appreciate the reviewer's suggestion and have added a concise explanation to Section 4.1 and a detailed illustration in Appendix A.4.
>
> In brief:
> 1.  **Flattening:** We maintain a separate KV cache per iteration depth. During attention, we flatten the 2D (token, depth) grid by concatenating deeper KV caches *after* shallower ones along the sequence dimension (Figure 14(b)).
> 2.  **Positional Encoding:** Each token retains its original 1D position index regardless of iteration depth. Iteration-aware LoRA designs naturally encode the depth information.
> 3.  **Masking:** The duo-causal mask over this flattened sequence enforces both temporal ($j \le i$) and depth-wise ($k \le d$) causality (Figure 14(c)).
>
> This design preserves compatibility with standard attention kernels and KV management systems.

---

### Official Review · Reviewer_8P1E · 2025-10-30

**Soundness:** 2
**Presentation:** 3
**Contribution:** 2
**Rating:** 4
**Confidence:** 5

**Summary:**

This paper proposes Think-at-Hard (TaH), a selective latent iteration method to address "latent overthinking" in Large Language Models (LLMs)—a flaw where uniform extra iterations corrupt correct predictions of easy tokens. TaH uses a neural decider to identify "hard tokens" (mispredicted in the first pass) for targeted latent refinement, with key designs including duo-causal attention (cross-iteration context flow), depth-specific LoRA adapters (objective shift support), and a two-stage training scheme (oracle-aligned backbone/decider training). Experiments on 5 mathematical reasoning benchmarks (GSM8K, MATH500, etc.) with Qwen3-0.6B/1.7B show 4.0–5.4% accuracy gains with <3% extra parameters, outperforming uniform baselines like AlwaysThink. However, the work suffers from weak baselines, limited model size validation, narrow domain coverage, and incomplete efficiency analysis—undermining its validity.

**Strengths:**

* The paper clearly defines "latent overthinking" (uniform iterations revising correct easy-token predictions) as a critical gap in prior latent reasoning methods (e.g., AlwaysThink). This insight directly addresses inefficiencies in existing designs and provides a clear motivation for selective iteration .
* The two-stage training scheme (backbone optimization via oracle policy first, then decider imitation on frozen backbone) decouples the tightly coupled backbone and decider, resolving distribution shift issues and achieving faster convergence (lower validation perplexity than Standard) .

**Weaknesses:**

1. TaH’s comparisons are limited to underpowered baselines that do not represent current state-of-the-art (SOTA) dynamic computation methods. The core baselines—Standard (vanilla Qwen3), AlwaysThink (uniform iterations), SoftThink (logits-weighted latent iterations), and simple routing (between small models)—lack comparisons with stronger alternatives: (1) MoR (Bae et al., 2025), a concurrent selective recursion method, is only claimed to require "complete retraining" without direct performance/efficiency comparisons on the same Qwen3 backbone or datasets ; (2) SOTA layer-skipping methods (e.g., MoD, FlexiDepth) are mentioned but not evaluated, leaving unclear how TaH outperforms dynamic compute methods beyond latent iteration ; (3) even when comparing with Ponder (a latent reasoning baseline), the paper uses a weak PonderingPythia-1.4B backbone (which scores only 2.0% on MATH500, far below Qwen3-0.6B’s 47.2%), making the comparison uninformative . These weak baselines mean TaH’s gains cannot be contextualized against real-world alternatives.
2. The paper only evaluates TaH on Qwen3-0.6B and 1.7B—two small-scale models (≤2B parameters)—with no validation on medium/large LLMs (e.g., 7B, 13B, 34B). This limitation is critical because: (1) small models have low performance ceilings; the 4.0–5.4% gains may stem from fixing basic prediction errors (e.g., logical connectives) rather than improving complex reasoning, which is the core goal of latent iteration ; (2) larger LLMs often exhibit different reasoning behaviors (e.g., more stable CoT generation) and may not suffer from "latent overthinking" to the same degree—TaH’s effectiveness on these models (where efficiency gains matter more for deployment) remains unproven ; (3) the paper highlights edge deployment (a key use case for small models) but does not test TaH on even smaller models (e.g., 360M MobileLLM-R1), limiting insights into its scalability for resource-constrained scenarios .
3. All core experiments are restricted to mathematical reasoning (GSM8K, MATH500, AMC23, etc.), with only a single, underdeveloped cross-domain test: a 1.7B TaH model trained on Open-R1’s science subset and evaluated on GPQA-diamond (a graduate-level Q&A benchmark). This experiment provides minimal evidence of domain agnosticism—non-mathematical tasks (e.g., code debugging, logic puzzles, scientific problem-solving) have distinct hard-token patterns (e.g., syntax vs. mathematical operators), and TaH’s decider may fail to identify hard tokens in these domains . Without broader validation, the claim that TaH is "domain-agnostic" is unsupported.
4. TaH’s oracle policy (used to label hard tokens) relies on a frozen SFT model’s top-1 prediction mismatch, but the paper never validates this labeling’s reliability: (1) it assumes the SFT model’s predictions are accurate enough to distinguish easy/hard tokens, but SFT models may misclassify easy tokens (e.g., simple coherence tokens like "the"), leading to unnecessary iterations or missed refinements ; (2) no comparison is made with alternative hardness metrics (e.g., token entropy, prediction cross-entropy), which are widely used in prior work—this leaves uncertainty about whether the oracle’s labels are optimal or consistent .

These weaknesses—especially weak baselines, small model size limits, and narrow domain validation—mean TaH’s contributions are not yet sufficiently robust for acceptance. The work would need to (1) compare with SOTA dynamic compute baselines, (2) validate on medium/large models, (3) expand to non-mathematical tasks, and (4) address oracle reliability and memory overhead to strengthen its case.

**Questions:**

See the weakness part.

---

> ### Author Response · Authors · 2025-11-22
>
> We sincerely thank Reviewer 8P1E for the extensive feedback.
> We appreciate your valuable recognition of our **clarity, insights, and training scheme design**. We address the concerns below.
>
> ### W1: Lack stronger baselines
>
> > TaH’s comparisons are limited to underpowered baselines that do not represent current state-of-the-art (SOTA) dynamic computation methods. The core baselines—Standard (vanilla Qwen3), AlwaysThink (uniform iterations), SoftThink (logits-weighted latent iterations), and simple routing (between small models)—lack comparisons with stronger alternatives:
>
> We thank the reviewer for suggesting related works. We add detailed discussions on MoR, MoD, FlexiDepth, and Ponder in the revised paper.
> Below, we provide direct comparisons where feasible and clarify the distinct advantages of TaH.
>
> #### **W1.1: Compare contemporaneous work MoR**
>
> > (1) MoR (Bae et al., 2025), a concurrent selective recursion method, is only claimed to require "complete retraining" without direct performance/efficiency comparisons on the same Qwen3 backbone or datasets;
>
> MoR requires pre-training from scratch, consuming $12-99\times$ more training tokens than TaH, which only requires fine-tuning.
> We provide a direct comparison by finetuning a TaH-360M model from MobileLLM-R1-360M-Base. We evaluated it over the official open-sourced MoR-360M (only size) on ARC-Easy and ARC-Challenge. Evaluation setups follow MoR's own protocol. As shown, TaH outperforms MoR on both benchmarks with much less re-training overhead.
>
> |Model|ARC-Easy|ARC-Challenge|
> |:--|--:|--:|
> |Standard (from MoR)|58.42|24.83|
> |MoR-Rec2-Expert|57.72|25.68|
> |MoR-Rec3-Expert|56.52|24.23|
> |TaH|59.81|28.00|
>
> We note that MoR is a contemporaneous work per ICLR review policy [1]. We acknowledge MoR's respective and complementary contributions with TaH.
>
> [1] “ICLR 2026 Reviewer Guide.” ICLR, 2026, iclr.cc/Conferences/2026/ReviewerGuide.
>
> #### **W1.2: Compare MoD and FlexiDepth**
>
> > (2) SOTA layer-skipping methods (e.g., MoD, FlexiDepth) are mentioned but not evaluated, leaving unclear how TaH outperforms dynamic compute methods beyond latent iteration;
>
> We clarify that layer-skipping and latent reasoning target different goals.
> MoD and FlexiDepth reduce FLOPs (computation) instead of parameter count (memory).
> In contrast, TaH targets parameter-constrained reasoning, enabling smaller models to reason better without increasing parameter size.
>
> Since direct comparison is impossible due to different base models (Llama3-8B vs Qwen-1.8B) and lack of official open code (MoD), we compare their reported relative gains:
>
> |Method|FlexiDepth|TaH+|
> |:--|:--|:--|
> |Base Model|Llama3|Qwen3|
> |Param.|8B|1.7B|
> |GSM8K Acc.|69.5%|84.5%|
> |Gain over Standard|+1.6%|**+3.7%**|
>
> TaH achieves higher relative gains, validating its effectiveness. Again, we highlight their different objectives and complementary contributions.
>
> #### **W1.3: Compare with Ponder**
>
> > (3) even when comparing with Ponder (a latent reasoning baseline), the paper uses a weak PonderingPythia-1.4B backbone (which scores only 2.0% on MATH500, far below Qwen3-0.6B’s 47.2%),
>
> Ponder requires pre-training from scratch (~300B tokens), which is **>1000x** the token cost of TaH and exceeds standard academic resources. TaH is explicitly designed to unlock recurrent capabilities in *existing* pre-trained LLMs via efficient finetuning.
>
> Nevertheless, we add two comparisons:
> 1. **Direct Performance Comparison:** TaH achieves higher ARC accuracy over the official release version of Ponder, only using fewer parameters.
>
> |Method|Ponder|TaH+|
> |:--|:--|:--|
> |Base model|Pythia|Qwen3|
> |Param.|2.8B|1.8B|
> |ARC-Easy Acc.|70.6|90.2|
> |ARC-Challenge Acc.|35.8|85.1|
>
> 2. **Direct Architectural Ablation:** To isolate architectural benefits, we implemented Ponder's design (Fixed-depth + Standard Causal Attention) on Qwen3-0.6B. We tested it both in its original form and with our LoRA enhancement, following our ablation study setup.
>
> | Architecture | LoRA | Iteration Policy | Attention | MATH500 | AMC23 | OlympiadBench | Average |
> | :--- | :---: | :--- | :--- | ---: | ---: | ---: | ---: |
> | Ponder (Ours Impl.) | $\times$ | Fixed-depth | Causal (Current Iter) | 17.0 | 5.0 | 7.7 | 9.9 |
> | Ponder (with LoRA) | $\checkmark$ | Fixed-depth | Causal (Current Iter) | 44.6 | 20.6 | 16.0 | 27.1 |
> | **TaH** | **$\checkmark$** | **Dynamic** | **Duo-causal** | **51.2** | **32.5** | **23.9** | **35.9** |
>
> As shown, TaH's duo-causal attention, iteration decider, and LoRA designs significantly outperform the Ponder architectures when trained on the same data and Qwen3-0.6B-Base model, proving the superiority of our design choices.

---

> ### Author Response · Authors · 2025-11-22
>
> ### W2: Evaluate on larger LLMs
>
> > The paper only evaluates TaH on Qwen3-0.6B and 1.7B—two small-scale models (≤2B parameters)—with no validation on medium/large LLMs (e.g., 7B, 13B, 34B). This limitation is critical because:
> (1) small models have low performance ceilings; the 4.0–5.4% gains may stem from fixing basic prediction errors (e.g., logical connectives) rather than improving complex reasoning, which is the core goal of latent iteration ;
> (2) larger LLMs often exhibit different reasoning behaviors (e.g., more stable CoT generation) and may not suffer from "latent overthinking" to the same degree—TaH’s effectiveness on these models (where efficiency gains matter more for deployment) remains unproven ;
> (3) the paper highlights edge deployment (a key use case for small models) but does not test TaH on even smaller models (e.g., 360M MobileLLM-R1), limiting insights into its scalability for resource-constrained scenarios.
>
> We appreciate the reviewer's advice, and train TaH model of 4B size.
> The training follows the exact same setup as the main paper, except that we truncate data to 4K to avoid out-of-memory during training.
>
> |Method|AIME25|AIME24|OlympiadBench|AMC23|MATH500|GSM8K|Average|
> |:--|--:|--:|--:|--:|--:|--:|--:|
> |Standard|23.3|23.7|47.7|62.8|82.8|90.5|55.1|
> |SoftThink|22.5|20.0|50.1|64.1|83.2|90.9|55.1|
> |TaH|30.4|29.6|50.5|70.3|84.4|90.4|59.3|
>
> TaH-4B outperforms the Standard baseline by 4.2%, showing consistent gains over the 0.6B and 1.7B setups. Note that the accuracy gains on harder tasks are generally higher, showing that Think-at-Hard improves hard reasoning as expected.
>
> ### W3: Benchmark beyond math
>
> > All core experiments are restricted to mathematical reasoning (GSM8K, MATH500, AMC23, etc.), with only a single, underdeveloped cross-domain test: a 1.7B TaH model trained on Open-R1’s science subset and evaluated on GPQA-diamond (a graduate-level Q&A benchmark). This experiment provides minimal evidence of domain agnosticism—non-mathematical tasks (e.g., code debugging, logic puzzles, scientific problem-solving) have distinct hard-token patterns (e.g., syntax vs. mathematical operators), and TaH’s decider may fail to identify hard tokens in these domains. Without broader validation, the claim that TaH is "domain-agnostic" is unsupported.
>
> We agree that verifying TaH on broader domains is crucial. We address this in two ways:
>
> **1. Training on Diverse Tasks (General Reasoning & Code)**
> We expanded our evaluation to include general reasoning (GPQA, MMLU-STEM) and coding (HumanEval+, MBPP+).
> We followed the same setups of the paper to finetune Qwen3-1.7B-Base. The only change is to replace the math-only dataset with an equal sample of all task splits of OpenR1, totaling 100K data items, to maintain the same data scale as our main experiments.
>
> |Method|MATH500|AMC23|GPQA|MMLU-STEM|HumanEval+|MBPP+|Average|
> |:--|--:|--:|--:|--:|--:|--:|--:|
> |Standard|67.8|39.7|30.3|74.1|44.2|27.2|47.2|
> |TaH+|72.6|48.4|39.4|76.6|48.2|39.0|54.0|
> |Gain|+4.8|+8.7|+9.1|+2.5|+4.0|+11.8|+6.8|
>
> As shown, TaH achieves consistent gains across all domains, not just math.
> It demonstrates that TaH generalizes effectively to diverse reasoning and generation tasks.
>
> **2. Zero-Shot Out-of-Domain (OOD) Generalization**
> We also tested whether models trained *only* on math could directly generalize to general science tasks (MMLU-STEM) without further fine-tuning.
>
> |Benchmark|Standard|SoftThink|AlwaysThink|TaH+|
> |:--|--:|--:|--:|--:|
> |MMLU-STEM (0.6B)|51.6|51.4|42.6|**56.3**|
> |MMLU-STEM (1.7B)|70.8|70.6|63.8|**73.7**|
>
> These results show that TaH consistently outperforms baselines even on OOD tasks, suggesting TaH captures generalizable reasoning patterns rather than memorizing domain-specific shortcuts.
>
> Detailed setups and results have been added to Section 5.2 and Appendix A.2.1.

---

> ### Author Response · Authors · 2025-11-22
>
> ### W4.1: Verify oracle policy reliability
>
> > TaH’s oracle policy (used to label hard tokens) relies on a frozen SFT model’s top-1 prediction mismatch, but the paper never validates this labeling’s reliability: (1) it assumes the SFT model’s predictions are accurate enough to distinguish easy/hard tokens, but SFT models may misclassify easy tokens (e.g., simple coherence tokens like "the"), leading to unnecessary iterations or missed refinements;
>
> We wish to clarify oracle policy robustness from two perspectives.
>
> **1. Reference policy objective.** The reference model acts as a failure detector, not a teacher.
> Therefore, "hardness" in our context does not need to align with human perception or absolute difficulty; it simply marks model-specific failures.
> If the reference model is biased or inaccurate, it exposes errors; TaH flags them as hard, and precisely targets these tokens for correction with latent iterations.
> Since the reference model and TaH backbone share the same base model, the reference bias naturally directs TaH to the specific areas where the model needs refinement.
>
> **2. Empirical Verification** Experiments confirm the robustness of oracle policy in (1) stabilizing and improving backbone *training* (Table 3) and (2) raising performance ceiling in *inference*, even for fixed-depth-trained models (Table 4).
> We further show that even a smaller, "inaccurate" reference model (1.7B) successfully identifies 81% of a larger (4B) model's hard tokens (Appendix A.3.1). The identified tokens correspond to the 4B model's highest uncertainty ($\approx 2.0\times$ higher loss), confirming reliable hardness detection.
>
> ### W4.2: Alternative hardness metrics
>
> > (2) no comparison is made with alternative hardness metrics (e.g., token entropy, prediction cross-entropy), which are widely used in prior work—this leaves uncertainty about whether the oracle’s labels are optimal or consistent .
>
> We appreciate the discussion and explore the mentioned token entropy and prediction cross-entropy. We examine their influence on training dynamics and final results using 0.6B models.
> The loss curves and detailed setups are added to Appendix A.3.1.
>
> |Method|MATH500|AMC23|Olympiadbench|Average|
> |:--|--:|--:|--:|--:|
> |TaH-Entropy|42.0|21.9|16.9|26.9|
> |TaH-CE|47.4|21.2|20.4|29.7|
> |TaH|51.2|32.5|23.9|35.9|
>
> As shown in the table, the original Top-1 hardness criteria of TaH achieves the highest performance.

---

### Official Review · Reviewer_Q1jn · 2025-10-31

**Soundness:** 3
**Presentation:** 2
**Contribution:** 3
**Rating:** 6
**Confidence:** 3

**Summary:**

This paper introduces Think-at-Hard (TaH), a method that selectively applies latent iterations only to hard tokens in language model reasoning tasks, rather than applying uniform multiple iterations to all tokens as in prior approaches. TaH applies LoRA modules and residual connections to the LLM backbone to better facilitate such objective shift. It also design a duo-causal attention mechanism for cross-depth contextualization. Experiments on five math-focused benchmarks (GSM8K, MATH500, AMC23, AIME25, OlympiadBench) show the effectiveness of the proposed TaH.

**Strengths:**

1. The idea that from latent overthinking to think-at-hard is reasonable, and the experiments show the good improvement.
2. The paper is well-written, making it understandable and easy to follow. It has a well-organized structure, and the proposed method is presented clearly.
3. The figures are well constructed, visually appealing, and help to understand the proposed methodology.

**Weaknesses:**

1.The core mechanism depends on an oracle policy π derived from a “frozen reference LLM” (the SFT variant of the base model) to label “hard” tokens (Eq. 7). This raises two problems: 1) Robustness: the “hardness” signal is tightly coupled to a particular reference model and training distribution; label quality and transfer to new domains are unclear. 2) Generalization: Main results are math-dominated; outside of a brief science subset note in the appendix, there is limited main-paper evidence that the oracle-imitation decider works for non-math reasoning (commonsense, code, multimodal). The method’s central promise that token-wise adaptive compute needs broader validation to justify venue-level impact.

2. In table 1, TaH surpass TaH+ (unpruned) in some benchmark (0.6B: AMC23, 1.7B: MATH500, AIME25). So it is hard to isolate whether gains come from TaH versus architectural changes in depth/capacity.

3. In Table 2, removing LoRA sometimes improves a task (MATH500 51.6 vs 51.2), and overall benefits fluctuate per dataset; this weakens the case that depth-specific adapters are consistently necessary.

**Questions:**

1. Could you report main-paper results on at least one non-math benchmark (e.g., GPQA-diamond in the main tables, CommonsenseQA, code tasks) with the same training setup, including ablations?
2. Could you provide a Standard-pruned baseline with the same depth as TaH (after pruning) to isolate TaH’s effect?
3. Please report latency/throughput/memory under realistic batching for Standard vs. AlwaysThink vs. TaH on the same hardware; include cache behavior with duo-causal masking.

---

> ### Author Response · Authors · 2025-11-22
>
> We sincerely thank Reviewer Q1jn for the positive feedback.
> We appreciate your valuable recognition of our **idea, experimental results and presentation**. We address the concerns below.
>
> ### W1.1.1: Oracle policy robustness
>
> > The core mechanism depends on an oracle policy π derived from a “frozen reference LLM” (the SFT variant of the base model) to label “hard” tokens (Eq. 7). This raises two problems:
> >
> > 1) Robustness: the “hardness” signal is tightly coupled to a particular reference model and training distribution;
>
> We wish to clarify oracle policy robustness from two perspectives.
>
> **Reference policy objective.** The reference model acts as a failure detector, not a teacher.
> Therefore, "hardness" in our context does not need to align with human perception or absolute difficulty; it simply marks model-specific failures.
> If the reference model is biased or inaccurate, it exposes errors; TaH flags them as hard, and precisely targets these tokens for correction with latent iterations.
> Since the reference model and TaH backbone share the same base model, the reference bias naturally directs TaH to the specific areas where the model needs refinement.
>
> **Empirical Verification** Experiments confirm the robustness of oracle policy in (1) stabilizing and improving backbone *training* (Table 3) and (2) raising performance ceiling in *inference*, even for fixed-depth-trained models (Table 4).
> We further show that even a smaller, "inaccurate" reference model (1.7B) successfully identifies 81% of a larger (4B) model's hard tokens (Appendix A.3.1). The identified tokens correspond to the 4B model's highest uncertainty ($\approx 2.0\times$ higher loss), confirming reliable hardness detection.
>
> ### W1.1.2: Iteration decider generalizability
>
> > label quality and transfer to new domains are unclear.
>
> The trained iteration decider can generalize well to general domains.
>
> **Decider's empirical generalizability.**
> We evaluated the iteration decider (trained on general OpenR1) across three distinct validation subsets: math, code, and general QA.
>
> |       | math  | code  | QA |
> | ----- | ----- | ----- | -------- |
> | Iter. Percentage | 7.8% | 10.7% | 26.6% |
> | Iter. Acc.  | 86.7% | 82.3% | 76.6%  |
> | Benchmark Gain over Standard | +6.8% | +7.9% | +5.8% |
>
> As shown in the table, the decider maintains high accuracy across all domains without retraining.
> Notably, the decider automatically adjusts the iteration rate (e.g., 26.6% for QA vs. 7.8% for Math) based on task difficulty, despite using the same 0.9 threshold.
> This confirms that the decider responds to intrinsic uncertainty signals rather than memorizing patterns, consistent with the token-level difficulty distributions observed in previous work [1].
>
> **Source of generalizability and robustness.**
> Why is the decider generalizable and robust? We investigate and find that hardness classification is a relatively simple task with clear, task-agnostic signals.
> In Appendix A.3.2, we show that "hard" tokens share a universal signature across domains: significantly higher entropy (>5x) in the first-pass logit distribution. This confirms that "hardness" is an robustly identifiable property, a task so distinct it can be approximated even with simple entropy thresholds.
> Given this, the neural decider easily learns reliable classification strategies that generalize well.
>
> [1]Fu, Tianyu, et al. “R2R: Efficiently Navigating Divergent Reasoning Paths with Small-Large Model Token Routing.” NeurIPS 2025

---

> ### Author Response · Authors · 2025-11-22
>
> ### W1.2, Q1: Benchmark beyond math
>
> > W1.2: 2) Generalization: Main results are math-dominated; outside of a brief science subset note in the appendix, there is limited main-paper evidence that the oracle-imitation decider works for non-math reasoning (commonsense, code, multimodal). The method’s central promise that token-wise adaptive compute needs broader validation to justify venue-level impact.
> >
> > Q1: Could you report main-paper results on at least one non-math benchmark (e.g., GPQA-diamond in the main tables, CommonsenseQA, code tasks) with the same training setup, including ablations?
>
> We agree that verifying TaH on broader domains is crucial. We address this in two ways:
>
> **1. Training on Diverse Tasks (General Reasoning & Code).**
> We expanded our evaluation to include general reasoning (GPQA, MMLU-STEM) and coding (HumanEval+, MBPP+).
> We followed the same setups of the paper to finetune Qwen3-1.7B-Base. The only change is to replace the math-only dataset with a equal sample of all task splits of OpenR1, totaling 100K data items to maintain the same data scale as our main experiments.
>
> |Method|MATH500|AMC23|GPQA|MMLU-STEM|HumanEval+|MBPP+|Average|
> |:--|--:|--:|--:|--:|--:|--:|--:|
> |Standard|67.8|39.7|30.3|74.1|44.2|27.2|47.2|
> |TaH+|72.6|48.4|39.4|76.6|48.2|39.0|54.0|
> |Gain|+4.8|+8.7|+9.1|+2.5|+4.0|+11.8|+6.8|
>
> As shown, TaH achieves consistent gains across all domains, not just math.
> It demonstrates that TaH generalizes effectively to diverse reasoning and generation tasks.
>
> **2. Zero-Shot Out-of-Domain (OOD) Generalization.**
> We also tested whether models trained *only* on math could directly generalize to general science tasks (MMLU-STEM) without further fine-tuning.
>
> |Benchmark|Standard|SoftThink|AlwaysThink|TaH+|
> |:--|--:|--:|--:|--:|
> |MMLU-STEM (0.6B)|51.6|51.4|42.6|**56.3**|
> |MMLU-STEM (1.7B)|70.8|70.6|63.8|**73.7**|
>
> These results show that TaH consistently outperforms baselines even on OOD tasks, suggesting TaH captures generalizable reasoning patterns rather than memorizing domain-specific shortcuts.
>
> Detailed setups and results have been added to Section 5.2 and Appendix A.2.1.
>
> ### W2, Q2: Compare standard-pruned performance
>
> > W2: In table 1, TaH surpass TaH+ (unpruned) in some benchmark (0.6B: AMC23, 1.7B: MATH500, AIME25). So it is hard to isolate whether gains come from TaH versus architectural changes in depth/capacity.
> >
> > Q2: Could you provide a Standard-pruned baseline with the same depth as TaH (after pruning) to isolate TaH’s effect?
>
> We add the requested Standard-Pruned baseline to address the concern.
> This baseline is Qwen2.5-1.7B model with the same layers removed as in our TaH configuration.
> As shown in the table below, simply pruning layers degrades performance (avg. 46.6% vs 47.8%).
> This confirms that TaH's gains come from its architectural designs, not from reduced depth or parameter count.
>
> |Method|AIME25|OlympiadBench|AMC23|MATH500|GSM8K|Average|
> |:--|--:|--:|--:|--:|--:|--:|
> |Standard|13.3|33.0|42.2|68.4|82.1|47.8|
> |Standard-Pruned|11.7|32.7|41.3|68.0|79.4|46.6|
>
> We note that the minor variations where TaH occasionally surpasses TaH+ are within the margin of variance typical for generation tasks.
> We use 8 rollouts per question for small benchmarks (see Section 5.1), trying to minimize noise within our resource budget.
> Crucially, both TaH and TaH+ consistently and noticeably outperform other baselines, validating the method's effectiveness regardless of the specific pruning configurations.
>
> ### W3: Necessity of LoRA
>
> > In Table 2, removing LoRA sometimes improves a task (MATH500 51.6 vs 51.2), and overall benefits fluctuate per dataset; this weakens the case that depth-specific adapters are consistently necessary.
>
> We doubled the rollout count for MATH500 and OlympiadBench to obtain more stable estimates, in response to the concern about performance fluctuations. We report the mean accuracy $\pm$ sample standard deviation below.
>
> |LoRA|Residual|Attention|MATH500|AMC23|Olympiadbench|Average|
> |:--|:--|:--|--:|--:|--:|--:|
> |✓|✓|Duo-causal|$52.1 \pm 1.3$|$32.5 \pm 1.8$|$23.9 \pm 2.3$|36.2|
> |✗|✓|Duo-causal|$51.6 \pm 0.0$|$29.7 \pm 1.8$|$22.4 \pm 1.5$|34.6|
>
> With reduced variance, depth-specific LoRA adapters consistently improve performance (36.2% vs 34.6% avg). The original fluctuation on MATH500 falls within the margin of variance, while gains on harder tasks (AMC23 and OlympiadBench) are noticeable, confirming the necessity of depth-specific parameters.

---

> ### Author Response · Authors · 2025-11-22
>
> ### Q3: Report latency, throughput, and memory
>
> > Please report latency/throughput/memory under realistic batching for Standard vs. AlwaysThink vs. TaH on the same hardware; includes cache behavior with duo-causal masking.
>
> We report efficiency metrics on a single NVIDIA A800-80GB GPU (1.7B model, 8K output length). We analyze both empirical results and theoretical costs (FLOPs/Memory Access).
> Details are in Appendix A.2.2 and A.2.3, we brief results here.
>
>
> | Method | Memory (GB) | Latency (s) | Throughput (tok/s) | *Breakdown (s)* | | | |
> | :--- | ---: | ---: | ---: | ---: | ---: | ---: | ---: |
> | | | | | **Iter-1** | **Iter-2** | **Decider** | **LoRA Switching** |
> | Standard | 4.3 | 210.6 | 38.9 | 210.6 | – | – | – |
> | AlwaysThink | 6.8 | 747.2 | 11.0 | 224.1 | 384.7 | – | 91.1 |
> | TaH | 4.6 | 301.4 | 27.2 | 229.8 | 29.6 | 10.5 | 7.5 |
>
> | Method | Normed FLOPs | Normed Mem Access |
> | :--- | ---: | ---: |
> | Standard | 1.00x | 1.00x |
> | AlwaysThink | 2.19x | 2.19x |
> | TaH | 1.05x | 1.05x |
>
>
> **1. TaH vs. AlwaysThink.**
> TaH averages only **1.06 iterations per token**, drastically reducing the cost compared to the fixed 2.00 iterations of AlwaysThink.
> Theoretically, this yields a $2.09\times$ reduction in FLOPs and memory access. Empirically, TaH achieves a **$2.48 \times$ speedup** and **$1.48\times$ peak memory reduction** over AlwaysThink, proving the efficiency of our selective strategy.
>
> **2. TaH vs. Standard.**
> Theoretically, TaH incurs only marginal overhead (**1.05$\times$**) over Standard.
> The current empirical latency gap (1.43$\times$) is primarily driven by unoptimized engineering overheads, such as Python-level LoRA switching.
> The engineering optimizations are orthogonal to the algorithmic contribution of TaH, and we will continue to refine the system to further reduce latency and memory overhead.

---

### Official Review · Reviewer_yXXU · 2025-10-31

**Soundness:** 3
**Presentation:** 3
**Contribution:** 3
**Rating:** 6
**Confidence:** 4

**Summary:**

This paper proposes TaH, a selective latent iteration framework that applies deeper internal reasoning only to $hard$ tokens, which are initially incorrectly predicted by a reference model, and directly generating $easy$ tokens. The method introduces a duo-causal attention mechanism to enable cross-iteration information flow while preserving training parallelism. It also uses LoRA adapters for refinement in deeper iterations. For training, it employs a two-stage training scheme: first training the backbone under a static oracle policy derived from a supervised fine-tuned (SFT) reference model, then training a lightweight neural decider to imitate this policy. Experiments on five reasoning benchmarks show consistent improvements over strong baselines with minimal computational overhead.

**Strengths:**

1. The paper is well-motivated. It identifies the issue of latent overthinking, and proposes a selective method for solution.
2. The duo-causal attention mechanism is proposed to enable depth-wise information flow. The two-stage decoupled training is also reasonable.
3. The authors conduct extensive studies and validate the method’s performance.
4. The paper is overall well-written and easy to follow.

**Weaknesses:**

Though the paper is overall reasonable to me, there may exist several potential weaknesses.
1. The proposed method seems to heavily rely on the supervised oracle policy. The oracle policy depends on ground-truth tokens and a frozen SFT reference model, limiting applicability to settings without high-quality supervision. It also remains unclear how TaH would perform if the reference model itself is biased or inaccurate on certain hard examples.
2. The experiments are mostly conducted on mathematical reasoning. The method’s effectiveness on more diverse tasks, such as commonsense reasoning, dialogue, or code generation, where $hardness$ is less local and more context-dependent, is not demonstrated.
3. There may exist potential robustness issue in the decider. The decider is trained in a specific data domain. It may produce less effective behavior on out-of-distribution or adversarial inputs. A single misclassification may permanently forfeit refinement opportunity, unlike AlwaysThink which provides a safety strategy.

**Questions:**

Please see strengths and weaknesses.

Besides, it would be beneficial to further clarify how the approach differs from other adaptive-computation methods in the latent reasoning literature, as a discussion in the paper.

---

> ### Author Response · Authors · 2025-11-22
>
> We sincerely thank Reviewer yXXU for the positive feedback.
> We appreciate your valuable recognition of our **motivation, method design, extensive experiments and writing**. We address the concerns below.
>
> ### W1: Reliance on oracle policy
>
> > The proposed method seems to heavily rely on the supervised oracle policy. The oracle policy depends on ground-truth tokens and a frozen SFT reference model, limiting applicability to settings without high-quality supervision. It also remains unclear how TaH would perform if the reference model itself is biased or inaccurate on certain hard examples.
>
> We appreciate the reviewer's concern and clarify the supervision requirements, objective, adding empirical results and analysis.
>
> **Data requirement.** The oracle policy does not impose additional constraints on data quality over standard Supervised Fine-Tuning (SFT).
> All baselines in our paper (Standard, AlwaysThink, TaH, TaH+) use the exact same SFT dataset (OpenR1). TaH's superior performance shows its ability to utilize existing data more effectively, rather than relying on higher-quality ground truth.
> The reference model is simply a *standard SFT-trained variant* of the same base LLM, which is readily available in standard training pipelines with checkpoints after the SFT stage.
>
> **Reference policy objective.** The reference model acts as a failure detector, not a teacher.
> Therefore, "hardness" in our context does not need to align with human perception or absolute difficulty; it simply marks model-specific failures.
> If the reference model is biased or inaccurate, it exposes errors; TaH flags them as hard, and precisely targets these tokens for correction with latent iterations.
> Since the reference model and TaH backbone share the same base model, the reference bias naturally directs TaH to the specific areas where the model needs refinement.
>
> **Empirical Verification** Experiments confirm the robustness of oracle policy in (1) stabilizing and improving backbone *training* (Table 3) and (2) raising performance ceiling in *inference*, even for fixed-depth-trained models (Table 4).
> We further show that even a smaller, "inaccurate" reference model (1.7B) successfully identifies 81% of a larger (4B) model's hard tokens (Appendix A.3.1). The identified tokens correspond to the 4B model's highest uncertainty ($\approx 2.0\times$ higher loss), confirming reliable hardness detection.
>
> ### W2: Benchmark beyond math
>
> > The experiments are mostly conducted on mathematical reasoning. The method’s effectiveness on more diverse tasks, such as commonsense reasoning, dialogue, or code generation, where hardness is less local and more context-dependent, is not demonstrated.
>
> We agree that verifying TaH on broader domains is crucial. We address this in two ways:
>
> **1. Training on Diverse Tasks (General Reasoning & Code)**
> We expanded our evaluation to include general reasoning (GPQA, MMLU-STEM) and coding (HumanEval+, MBPP+).
> We followed the same setups of the paper to finetune Qwen3-1.7B-Base. The only change is to replace the math-only dataset with a equal sample of all task splits of OpenR1, totaling 100K data items to maintain the same data scale as our main experiments.
>
> |Method|MATH500|AMC23|GPQA|MMLU-STEM|HumanEval+|MBPP+|Average|
> |:--|--:|--:|--:|--:|--:|--:|--:|
> |Standard|67.8|39.7|30.3|74.1|44.2|27.2|47.2|
> |TaH+|72.6|48.4|39.4|76.6|48.2|39.0|54.0|
> |Gain|+4.8|+8.7|+9.1|+2.5|+4.0|+11.8|+6.8|
>
> As shown, TaH achieves consistent gains across all domains, not just math.
> It demonstrates that TaH generalizes effectively to diverse reasoning and generation tasks.
>
> **2. Zero-Shot Out-of-Domain (OOD) Generalization**
> We also tested whether models trained *only* on math could directly generalize to general science tasks (MMLU-STEM) without further fine-tuning.
>
> |Benchmark|Standard|SoftThink|AlwaysThink|TaH+|
> |:--|--:|--:|--:|--:|
> |MMLU-STEM (0.6B)|51.6|51.4|42.6|**56.3**|
> |MMLU-STEM (1.7B)|70.8|70.6|63.8|**73.7**|
>
> These results show that TaH consistently outperforms baselines even on OOD tasks, suggesting TaH captures generalizable reasoning patterns rather than memorizing domain-specific shortcuts.
>
> Detailed setups and results have been added to Section 5.2 and Appendix A.2.1.

---

> ### Author Response · Authors · 2025-11-22
>
> ### W3: Iteration decider robustness and generalizability
>
> > There may exist potential robustness issue in the decider. The decider is trained in a specific data domain. It may produce less effective behavior on out-of-distribution or adversarial inputs. A single misclassification may permanently forfeit refinement opportunity, unlike AlwaysThink which provides a safety strategy.
>
> We address the concern of iteration decider robustness and generalizability with empirical evidence and mechanism analysis. We also explain why imperfect iteration decider is better than AlwaysThink.
>
> **Decider's empirical generalizability.**
> We evaluated the iteration decider (trained on general OpenR1) across three distinct validation subsets: math, code, and general QA.
>
> |       | math  | code  | QA |
> | ----- | ----- | ----- | -------- |
> | Iter. Percentage | 7.8% | 10.7% | 26.6% |
> | Iter. Acc.  | 86.7% | 82.3% | 76.6%  |
> | Benchmark Gain over Standard | +6.8% | +7.9% | +5.8% |
>
> As shown in the table, the decider maintains high accuracy across all domains without retraining.
> Notably, the decider automatically adjusts the iteration rate (e.g., 26.6% for QA vs. 7.8% for Math) based on task difficulty, despite using the same 0.9 threshold.
> This confirms that the decider responds to intrinsic uncertainty signals rather than memorizing patterns, consistent with the token-level difficulty distributions observed in previous work [1].
>
> **Source of generalizability and robustness.**
> Why is the decider generalizable and robust? We investigate and find that hardness classification is a relatively simple task with clear, task-agnostic signals.
> In Appendix A.3.2, we show that "hard" tokens share a universal signature across domains: significantly higher entropy (>5x) in the first-pass logit distribution. This confirms that "hardness" is an robustly identifiable property, a task so distinct it can be approximated even with simple entropy thresholds.
> Given this, the neural decider easily learns reliable classification strategies that generalize well.
>
> **Comparison with AlwaysThink.**
> We respectfully disagree that AlwaysThink is always a safer strategy.
> Our results (Table 1, Table 4) show that AlwaysThink suffers from *latent overthinking*, degrading performance through false corrections on easy tokens.
> As shown in Figure 1, AlwaysThink risks degrading the 91% of tokens that are already correct at the first iteration to fix the minority.
> In contrast, the occasional decision errors of TaH are benign: if the decider misses a hard token, the model simply falls back to the standard single-pass prediction, which preserves strong performance thanks to TaH's depth-aware LoRA design.
> The table above confirms this empirically, showing TaH achieves consistent gains across tasks. The robustness to decision errors is further supported by TaH's robustness to the iteration threshold shown in Figure 4.
>
> [1] Fu, Tianyu, et al. “R2R: Efficiently Navigating Divergent Reasoning Paths with Small-Large Model Token Routing.” NeurIPS 2025
>
> ### Q1: Difference from other adaptive-computation methods
>
> > It would be beneficial to further clarify how the approach differs from other adaptive-computation methods in the latent reasoning literature, as a discussion in the paper.
>
> We sincerely thank the reviewer for the valuable advice. We have added the section in Section 2 and Appendix A.5.
>
> To the best of our knowledge, we are the first work to uncover the *latent overthinking* problem, highlighting the performance opportunity of dynamic latent iteration, in addition to its obvious efficiency benefits.
> Our duo-causal attention mechanism achieves both cross-iteration information flow and full sequence-level parallelism. The depth-aware LoRA and two-stage training also provide practical guidance to a powerful, stable dynamic latent reasoning model.
> We believe these methods form a unique insight and addition to the latent reasoning community.

---

### Author Response · Authors · 2025-12-02
**Summary of Key Points**

We thank reviewers and AC for the valuable feedback and time. We appreciate the consistent recognition of TaH’s **strong motivation** (all reviewers), **clear presentation** (yXXU, Q1jn), **sound method design** (yXXU, 8P1E, oeNK), and **consistent empirical gains** (yXXU, oeNK). Here we reiterate the key recognitions and updates during rebuttal.

---

### **Key recognitions**

**1. Strong motivation**

All reviewers unanimously acknowledge the significance of identifying the **latent overthinking** phenomenon (where fixed-depth iterations incorrectly revise easy predictions) and praise the proposed selective solution to address this critical gap.

**2. Clear presentation**

Reviewers find the paper to be **well-written**, easy to follow, and well-organized, with **high-quality figures** that effectively aid in understanding the methodology.

**3. Sound method design**

Reviewers recognize the technical novelty and effectiveness of the **duo-causal attention** and the decoupled **two-stage training scheme**, noting their success in enabling effective cross-depth information flow and fast, stable training convergence.

**4. Consistent empirical gains**

Reviewers acknowledge the extensive experiments across reasoning benchmarks, noting TaH's consistent accuracy gains of **4.0-11.3\%** over different fixed-depth methods, while **skipping latent iterations for 94% of tokens**.

---

### **Key rebuttal updates**

**1. Extended to general domains & code.** (Section 5.2, Appendix A.2.1)
> Addressing yXXU (W2), Q1jn (W1, Q1), 8P1E (W3)
   * **Diverse task training:** Extended experiments beyond math to general reasoning (GPQA, MMLU-STEM) and coding (HumanEval+, MBPP+), demonstrating consistent gains (+6.8% avg).
   * **OOD generalization:** Verified zero-shot Out-of-Domain (OOD) generalization on MMLU-STEM.

**2. Extended to larger models.** (Section 5).
> Addressing 8P1E (W2)
   * Validated TaH and TaH+ on a larger scale (4B model), showing consistent gains of 3.8% and 4.2% respectively.

**3. Comparisons with stronger baselines.** (Appendix A.5)
> Addressing 8P1E (W1)
   * Added direct comparisons with MoR, FlexiDepth, and Ponder; TaH consistently outperformed all added baselines.

**4. Oracle policy reliability.** (Appendix A.3.1)
>  Addressing yXXU (W1), Q1jn (W1), 8P1E (W4).
   * **Role clarification:** Clarified the reference model's role as a failure detector rather than a teacher, confirming robustness even if the reference is biased.
   * **Robustness verification:** A smaller reference model (1.7B) identifies 81% of hard tokens for a larger model (4B).
   * **Hardness validation:** Identified "hard" tokens indeed correspond to areas of high uncertainty (~2.0x higher loss) in the larger model.

**5. Iteration decider robustness.** (Appendix A.3.2)
> Addressing yXXU (W3), Q1jn (W1).
   * Validated the decider across Math, Code, and QA subsets. The decider automatically adapts iteration rates based on task difficulty and shows consistently high iteration accuracy.

**6. Design choice ablations.** (Section 5.3, Appendix A.3.1)
> Addressing Q1jn (W3, W2, Q2), 8P1E (W4), oeNK (W2, Q2)
   * **LoRA necessity:** Conducted multiple rollouts to reduce variance, confirming that LoRA adapters are essential for performance.
   * **Duo-causal necessity:** Ablated causal attention (attending to the current iteration) for TaH, confirming duo-causal's essential role for cross-depth information flow.
   * **Standard-pruned:** Added a parameter-matched "Standard-Pruned" baseline to rule out the influence of specific pruning configurations.
   * **Hardness metrics:** Compared with alternative hardness metrics (Entropy, Cross-Entropy) in Appendix A.3.1, confirming Top-1 mismatch yields the best performance.

**7. Efficiency analysis.** (Appendix A.2.2, A.2.3)
> Addressing Q1jn (W3, Q3).
   * Reported real-world latency, throughput, and memory usage. TaH achieves ~2.48x speedup and ~1.48x memory reduction compared to AlwaysThink.

**8. Implementation details.** (Appendix A.4.1)
> Addressing oeNK (W4)
   * Provided explanation and figure of the 2D token flattening and KV-cache management for duo-causal attention

**9. Design choice clarifications.**
> Addressing oeNK (W1, Q1)
   * **Different roles of decider and duo-causal attention**: The decider provides efficiency via hard selection, while duo-causal attention ensures accuracy via cross-depth information flow.

---

### Meta-Review · Area_Chair_hCgZ · 2026-01-18

**Summary:**

The paper proposes a post-training method to address the latent overthinking problems in small-scale LLMs by applying an adaptive number of latent iterations to each token according to its hardness (harder tokens need more iterations to be predicted correctly). The proposed method, "Think-at-Hard (TaH)", uses duo-causal attention to attend to the historical tokens from previous iterations, and uses LoRA to shift the training objective to latent reasoning. TaH adopts a two-stage training scheme: it first trains the backbone under a static oracle policy of latent iterations per token defined by the SFT reference model. Then it trains a lightweight neural decider with the backbone frozen to imitate this oracle policy. In experiments, TaH is used to finetune Qwen3-0.6B/1.7B models on Open-R1. Evaluations on five math reasoning benchmarks show consistent improvements over baselines with minimal computational overhead. In rebuttal, the authors extend the experiments to general reasoning and coding benchmarks, and Qwen3-4B.

**Reviewer Concerns:**

- The backbone models are two small-scale underdeveloped models (0.6B/1.7B), whose performance ceilings are specifically low on several benchmarks. The gain of the proposed method might be achieved by fixing some trivial errors in this case. Moreover, it is not clear whether larger models still suffer from such a latent overthinking problem and whether TaH is still effective on them.
- The oracle policy relies on a frozen small model (the base model's SFT version)’s top-1 prediction mismatch to define the hard tokens, whose quality can be questionable and unreliable. Moreover, the SFT version cannot faithfully reflect the hard tokens for the base model that needs to be finetuned.
- Other widely-used hardness metrics other than top-1 prediction mismatch need to be compared with.
- Experiments only focus on math reasoning tasks. It is unknown whether the method can be effective on other domains or generalize to other OOD tasks.
- Lack of comparison and discussion of several important baselines and models for latent reasoning.
- A clear explanation of the motivation and an ablation study regarding the functionality of duo-causal attention is not provided.
- Report and comparison of latency/throughput/memory are not available.
- It is not clear how LoRA and pruning contribute to the final gains claimed to be achieved by the proposed method.

**Reviewer Scores:**

- The initial review ratings are 4, 4, 6, 6, with confidence of 5, 4, 4, 3. No reviewers responded to the rebuttal or claimed to revise the original ratings.
- The authors took efforts to address the concerns of the reviewers and provided several additional groups of experiments in non-math reasoning domains, OOD generalization, a larger model (Qwen-4B), efficiency, etc. I found these results are helpful to address some major concerns.
- The authors also did a good job of explaining the complementary roles of iteration decider and duo-causal attention.
- That being said, most experiments, analyses, and ablation studies are still limited to the two small-scale, poorly performed models from the Qwen family. Hence, it is still not convincing that the observations and conclusions are general and useful to larger models from other model families.
- The proposed method is composed of several components and non-standard settings compared to other latent reasoning approaches. Based on the reported ablation study results, it is still not clear how much each component brings separately, and how they interfere with each other, making the final statements loose.
- Although I agree with the authors that the hardness focuses on the weakness of the model to be finetuned and does not need to be consistent with other models or human-perceived difficulty, the oracle policy's hard tokens are defined by the weaknesses of the SFT model instead of the base model (the one to be finetuned), which is still questionable.
- The paper aims to address a novel and potentially important problem for latent reasoning, and I encourage the authors to keep improving the submission based on the feedbacks.

---

### Decision · Program_Chairs · 2026-01-26

Reject